

# Importance of succession in estimating biomass loss: Combining remote sensing and individual-based forest models

Ulrike Hiltner[1,2,3], Andreas Huth[2,4,5], Rico Fischer[2]

5 [1]Institute of Geography, Friedrich-Alexander-University Erlangen-Nuremberg, Erlangen, 91058 Germany

[2]Department of Ecological Modelling, Helmholtz-Centre for Environmental Research GmbH - UFZ, 04318 Leipzig, Germany

[3]Department Environmental System Sciences, ETH Zurich, 8092 Zurich, Switzerland

[4] German Centre for Integrative Biodiversity Research – iDiv Halle-Jena-Leipzig, Leipzig, 04103 Leipzig, Germany

[5] Institute for Environmental Systems Research, University Osnabruck, Osnabruck, 49076 Osnabruck, Germany

10 *Correspondence to*: Ulrike Hiltner (u.hiltner.uh@gmail.com)

**Abstract**. Disturbances can have strong impacts on the dynamics and structure of tropical forests. They often lead to increased tree mortality and affect their behaviour as carbon sinks. In the future, the intensity of disturbances, such as extreme weather events, fires, floods, and biotic agents, will probably even increase, with more serious consequences for tropical forests than we have already observed. However, impacts of altering disturbances on rates of forest biomass loss through tree mortality (hereinafter: biomass mortality) have been little described yet. This complicates progress in quantifying the effects of climate change on forests globally.

This study aims to analyse the consequences of elevated tree mortality on forest dynamics and to provide a methodology that can reduce uncertainties in estimating biomass mortality rates at local and country level. We achieved this by linking benefits of individual-based forest modelling, statistical linear regression, and remote sensing. We applied an individual-based forest model to investigate the impact of varying disturbance regimes on the succession dynamic of a humid *Terra Firma* forest at the Paracou study site in French Guiana. By simulating increased tree mortality rates, we were able to investigate their influence on several forest attributes, namely biomass, leaf area index, forest height, gross primary production, net primary production, and biomass mortality. Based on simulations of leaf area index and forest height, we developed a linear multivariate regression model to project biomass mortality.

Our findings demonstrate that severe disturbances altered the succession pattern of the forests in favour of fast-growing species, which changed gross primary production, but net primary production remained stable. We also observed a strong influence on biomass mortality rates as well as observed complex relationships between these rates and single forest attributes (leaf area index, forest height, and biomass). By combining leaf area index and forest height we obtained relationships that allow an estimation of the biomass mortality. Based on these findings, we mapped the biomass mortality for whole French Guiana. We found a nation-wide biomass mortality of 3 % per year (standard deviation = 1.4 % per year).



The approach we describe here, provides a novel methodology for quantifying the spatio-temporal distribution of biomass loss, which has recently been identified as particularly critical for monitoring mortality hotspots. Quantifying biomass mortality rates may help reducing uncertainties in the terrestrial component of the global carbon cycle.

**Keywords**. FORMIND forest model, Paracou, French Guiana, basic mortality rate, biomass mortality map, biomass
residence time, carbon turnover time, tree mortality monitoring

### 1. Introduction

Tropical forests represent an important pool in the global carbon cycle as they store about 55 % of the terrestrial carbon in their living biomass ($471 \pm 93$ Pg C). They assimilate about $2.8 \pm 0.7$ Pg C per year (Pan et al., 2011). This carbon sink behaviour of tropical forests has considerably reduced the growth rate of atmospheric carbon dioxide (Friedlingstein et al.,
2019; Le Quéré et al., 2016). However, the assimilation capacity of carbon is affected by forest disturbances, which can cause rapid, extensive carbon loss (Pugh et al., 2019; Seidl et al., 2014). Increased tree mortality due to disturbances has been related to a reduction in the carbon sink of tropical forests (Brienen et al., 2015; Hubau et al., 2020). A number of recent studies discuss different climate-controlled mortality drivers, such as temperature (Clark et al., 2010), vapour pressure deficit (Trenberth et al., 2014), drought (Phillips et al., 2010), and wind throw (Marra et al., 2014). In addition, mechanical
disturbances may also lead to tree mortality, for instance, insect calamities (Coley and Kursar, 2014), fires (Brando et al., 2014; Slik et al., 2010), and lianas (Ingwell et al., 2010; Wright et al., 2015). An expected higher frequency and intensity of the mortality drivers, may result in an increase in tree mortality and associated physiological mechanisms (McDowell et al., 2018). This is a major risk to climate mitigation efforts (e.g. REDD+), because reductions in carbon assimilation rates of tropical forests could counteract attempts to compensate for climate change by protecting tropical forest ecosystems (Gumpenberger
et al., 2010; Le Page et al., 2013).

Mortality is a complex process, because disturbances leading to tree mortality can be diverse. Forest disturbances may be abrupt or continuous and have abiotic or biotic, allogenic or autogenic, as well as extrinsic or intrinsic causes (Franklin et al., 1987; McDowell et al., 2018). Furthermore, drivers of tree mortality often occur in combination, so the primary factors of death are not obvious (Franklin et al., 1987; McDowell et al., 2018). Tree mortality leads to temporal changes in stand structure,
tree species composition, and releases of resources, in particular biomass (Franklin et al., 1987). Consequently, tree death affects important forest demographic processes, e.g. tree growth and recruitment, which are influenced on the one hand by species-specific competition strategies (Snell et al., 2014) and on the other hand by environmental and competitive factors (e.g. light availability) (Kuptz et al., 2010; Poorter, 1999; Uriarte et al., 2004). The influence on forest demographic processes is determined by disturbance intensity, which can range from temporary loss of vitality to mortality (Kindig and Stoddart,
2003) of individual trees, forest stands, and entire landscapes. Finally, disturbances are heterogeneously distributed, so that tree mortality can be scattered or clustered (Franklin et al., 1987). Often, it is difficult to quantify tree mortality and to assess the consequences of alterations on forest dynamics and structure.



An approach to analyse impacts of disturbances is offered by individual-based forest modelling (Botkin et al., 1972; Bugmann, 2001; Bugmann et al., 2019; Shugart, 2002; Shugart et al., 2015). They are parameterised with forest inventory data and allow investigations of forest growth dynamics over longer periods. By simulating tree growth, regeneration, competition, and mortality for each tree, these models can contribute in estimating biomass gain and loss (Hiltner et al., 2018). By using an individual- and process-based approach, the spatial heterogeneity within forest stands can be well represented in such models (Fischer et al., 2016; Hiltner et al., 2018; Rasche et al., 2011; Shugart et al., 2018). To estimate carbon budgets of forest stands and entire landscapes, a combination of forest models and remote sensing is necessary (Rödig et al., 2017; Shugart et al., 2015). This combination may provide information on the spatial distribution of biomass loss due to tree mortality (hereafter: biomass mortality).

The aim of this study is to investigate the impact of increased tree mortality on forest dynamics and to provide a framework for estimating biomass mortality rates on local and country scales. Here, we address the following research questions in detail:

1. What are the consequences of increased tree mortality on the dynamics of several forest attributes (i.e., biomass, forest height, GPP, NPP, and LAI, biomass mortality rates) in tropical forests?
2. How can biomass mortality rates for disturbed and undisturbed tropical forests be estimated using remote sensing products?

We applied the forest model FORMIND (Fischer et al., 2016; Hiltner et al., 2018; Köhler and Huth, 2004) and analysed the succession patterns of several forest attributes, including aboveground biomass (hereafter: biomass), forest height, gross primary production (GPP), net primary production (NPP), leaf area index (LAI), and biomass mortality ($m_{AGB}$) under different disturbance levels at the Paracou study site in French Guiana. The studied forests are characteristic for the country (Guitet et al., 2018), whose *Terra Firma* rainforests are generally dense and species rich, with 150–200 tree species per hectare (Gourlet-Fleury et al., 2004; Grau et al., 2017; Piponiot et al., 2016a). We simulated long-term forest development in a set of scenarios with different disturbance intensities and compared the model outputs with a reference scenario reflecting natural forest growth. The reference scenario is based on a previous study, in which tree size distribution, functional species composition and biomass were compared with records from forest inventories (35 a, 65 ha) (Hiltner et al., 2018). We propose that biomass mortality is a function of competition for light, disturbance intensity, and the successional stage of the forest. For up-scaling rates of biomass mortality from stand to landscape level, we used simulated forest height and LAI as proxies for successional stage, disturbance intensity, and light availability. Therefore, remote sensing-derived maps of tree height (Simard et al., 2011) and LAI (Myneni et al., 2015) were linked with the individual-based forest model.



## 2.    Materials and Methods

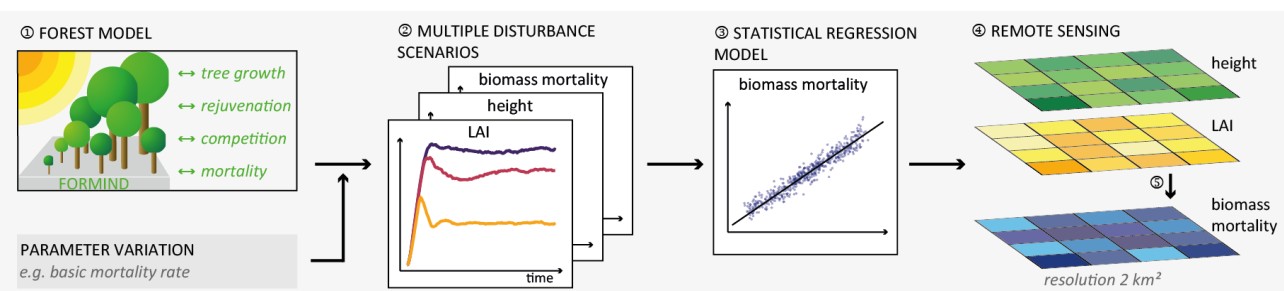

**Figure 1: Five work steps for deriving regional biomass mortality rates by linking a forest model and remote sensing. 1. Use a forest model to reproduce the natural succession dynamics locally as a reference. 2. Run the model to simulate the forest attributes leaf area index, forest height, and biomass mortality rate in a set of different disturbance scenarios. 3. Fit a regression model to the results of the simulation runs. 4. Apply the regression model to each pixel of the regional maps containing the values of the two forest attributes (here: leaf area index and forest height). 5. Derive a biomass mortality map. (Biomass mortality: rate of biomass loss due to tree mortality, LAI: leaf area index, height: forest height)**

### 2.1    Study region

The study site Paracou (location: -52.923793°, 5.274018°) is located in French Guiana, 95 % of which is covered by humid lowland *Terra Firma* forests (Hammond, 2005; Stach et al., 2009). These forests are characteristic for the Guiana Shield (Grau et al., 2017). They are generally species-rich with an average of 150–200 tree species per hectare (Gourlet-Fleury et al., 2004) and are dense in biomass stock (Rödig et al., 2017; Saatchi et al., 2011). We assume that the Paracou forest stand analysed and simulated by Hiltner et al. (2018) is representative for the whole region of *Terra Firma* forests in French Guiana.

### 2.2    Forest model FORMIND

#### 2.2.1    Model description

Our assessments were based on the Paracou version (Hiltner et al., 2018) of the forest model FORMIND v3.2 (Fischer et al., 2016) to analyse the forest dynamics under impacts of different levels of disturbance. FORMIND is an individual-based forest model that describes the vertical and horizontal forest structure and forest dynamics in squared patches of 400 m$^2$. The entire simulation area (here: 16 ha) consists of those patches.

Based on forest inventory data, every tree with a stem diameter at breast height (DBH) ≥ 0.1 m was simulated considering the following main processes at annual time steps: tree growth, regeneration, mortality, and competition for light and space. The biomass gain of a tree resulted from the difference between photosynthetic production and respiratory losses (Fischer et al., 2016; Hiltner et al., 2018). In the model, tree mortality is a key driver of forest dynamics. Tree mortality increased, if the space for canopy expansion was limited depending on a tree's position within the forest stand (self-thinning by crowding), if tree growth was reduced (growth-dependent), and surrounding trees could die after large trees had fallen (gap formation). Finally, each tree was subject to a basic mortality rate, which is stochastic. Here, we modified the basic mortality rate to assess different





disturbance intensities. Possible factors altering mortality rates include environmental drivers, such as extreme climate events, forest fires, wind-throw, and disease. Disturbance intensity was defined as a basic mortality rate of individual trees at stand
level with an arbitrarily distributed spatial disturbance pattern.

According to Hiltner et al. (2018), all tree species at Paracou were classified into plant function types (PFT) according to species-specific traits, i.e. maximum increment rates of DBH and maximum tree height. We simulated PFTs grouped into three successional types according to their light demand: light-requiring pioneer species, species with intermediate light requirements, and shade-tolerant climax species (Hiltner et al., 2018). A detailed model description can be found in Fischer et
al. (2016) and Hiltner et al. (2018) and online at www.formind.org.

### 2.2.2 Simulation setting

In order to investigate the effects of different disturbance levels on the dynamics of various forest attributes, we developed nine simulation scenarios with varying basic mortality (Fig. 1): One baseline scenario representing the natural forest development at Paracou with a PFT-specific ($p$) basic mortality rate ($m_p$), as well as eight scenarios differing in $m_p$. To obtain
$m_p$, the baseline's basic mortality rate per PFT ($m_{p \cdot bl}$) was multiplied by a factor ($f$) for each scenario ($sc$) so that:

$$m_{p,sc} = f \cdot m_{p,bl}, \qquad \text{with } f \in \left\{ \tfrac{1}{5}, \tfrac{1}{4}, \tfrac{1}{3}, \tfrac{1}{2}, 2, 3, 4, 5 \right\} \quad (1).$$

This resulted in different forest disturbance intensities on stand level per simulated scenario (Tab. 1). The scenario with $f = 1$, represented the baseline scenario. From this scenario, simulation results of biomass and tree size distribution at equilibrium were thoroughly compared with forest inventory data (over 35 years and 65 ha) by Hiltner et al. (2018).

All simulations used an annual time step that started in year 0 on bare ground and ended after 300 years. In the baseline scenario a forest stand reached its equilibrium after 210 years.

From the model output, we analysed the development of multiple forest attributes such as aboveground biomass (AGB), LAI, and forest height (mean height of the tallest three trees per 40 m x 40 m; Rödig et al., 2017; Simard et al., 2011), gross primary production (GPP), net primary production (NPP), and rate of biomass loss due to tree mortality ($m_{AGB}$) which we defined as
proportion of dead biomass ($AGB_{dead}$) to total AGB ($AGB_{total}$):

$$m_{AGB} = AGB_{dead} \cdot AGB_{total}^{-1} \quad (2).$$

In addition, we have computed the time period until each forest attribute has reached the stable state (hereafter: equilibrium time) as well as the mean biomass residence times ($\tau$), with $\tau$ averaged over all successional stages (simulated years 0–300). According to Carvalhais et al. (2014), $\tau$ can be defined as the ratio of biomass stock and outflux of biomass. However, biomass
outflux is not yet observable over large spatial scales (Thurner et al., 2016). Therefore, it was defined that biomass outflux equals biomass influx for forests in equilibrium (Carvalhais et al., 2014). Transferred to our study, the stock corresponds to





the total biomass ($AGB_{total}$), influx to GPP, and outflux to dead biomass ($AGB_{dead}$). Therefore, the following holds true for forests in equilibrium:

$$\tau = stock \cdot flux^{-1} = AGB_{total} \cdot GPP^{-1} = AGB_{total} \cdot AGB_{dead}^{-1} \qquad (3).$$

To compare the approach of Carvalhais et al. (2014) with this study, $\tau$ can be calculated from equations 2 and 3 as the reciprocal of the biomass mortality rate:

$$\tau = 1 \cdot m_{AGB}^{-1} \qquad \text{for } m_{AGB} > 0 \qquad (4).$$

Using equation 4, we calculated $\tau$ by taking forest succession into account.

**Table 1: Average forest disturbance intensities per simulation scenario plus specification (see eq. 1).**

| Factor $f$ | Average disturbance intensity $m_{p,sc}$ (a$^{-1}$) | Specification |
|---|---|---|
| 1 | 0.0129 | Baseline |
| 1/5 | 0.00258 | Low impact |
| 1/4 | 0.003225 | |
| 1/3 | 0.0042957 | |
| 1/2 | 0.00645 | |
| 2 | 0.0258 | |
| 3 | 0.0387 | |
| 4 | 0.0516 | |
| 5 | 0.0645 | High impact |

**2.3   Estimation of biomass mortality from remote sensing data**

To estimate the rate of biomass mortality, we decided to work with common forest attributes, which are available as remote sensing products. All training data of the forest attributes LAI and forest height were obtained as model outputs of the simulation experiments with the FORMIND forest model for the Paracou site (Fig. 1; cf. chap. 2.2). We assume that the rate of biomass mortality depends on the successional stage of the forests and disturbance level. We tested different statistical

models for combinations of the proxy variables LAI and forest height. The best guess is a multivariate linear model, which was used to describe variations in $m_{AGB}$ as a function of the two proxy variables.

We estimated $m_{AGB}$ [a$^{-1}$] as follows:

$$m_{AGB} = \beta_H \cdot H + \beta_L \cdot L + \varepsilon \qquad (5),$$

where $H$ is the forest height [m], $L$ the LAI, $\varepsilon$ the error term, and $\beta_i$ are the regression coefficients of the $i^{th}$ forest attribute.

The intercept was set to 0, as the biomass mortality rate is expected to be 0 when both LAI and forest height equal 0.





## 2.4 Establishment of the biomass mortality map

### 2.4.1 Input maps

To estimate forest height, we used a global map in the geographical projection WGS-84 with approximately 1 km pixel size
(Simard et al., 2011; Fig. S3.a). To create a LAI map, we used data from the MCD15A2H Version 6 Moderate Resolution
Imaging Spectroradiometer (MODIS) Level 4 with a pixel size of 500 m, and averaged the LAI values between 2005-05-20
and 2005-06-23 (Myneni et al., 2015). We harmonised and stacked the two input maps by first projecting the LAI map onto
the coordinate reference system of the forest height map using the Geospatial Data Abstraction Library for French Guiana
(www.gdal.org). The resampling was conducted with the bilinear method. The spatial aggregation of the LAI map (Fig. S3.b)
was performed by calculating the mean value of pixels whose centre lay within a 1-km cell of the forest height map.

### 2.4.2 Output map

The biomass mortality rates of French Guiana were estimated for each pixel by applying the multivariate linear regression
model (eq. 5 and 6) to the two input maps (Fig. 1). The biomass mortality values were then averaged over a pixel size of 2
km$^2$. Our regression model estimated negative biomass mortality rates for a small portion of pixels, which were excluded from
the biomass mortality map. This was mainly the case for pixels without forest cover according to a land use map published by
Stach et al. (2009). Please refer to supplements Tab. S1 for the computer software used in this study.

## 3 Results

### 3.1 Influence of increased tree mortality on the forest succession dynamics

In order to analyse the influence of varying disturbance intensities, we simulated the succession dynamics, which was affected
by competition between individual trees (succession types; Fig. 2). Here, we show that four successional phases can be
differentiated based on the development of the total stand biomass (Fig. 2). After 40 years of forest succession, the stand
biomass peaked at 500 $t_{ODM}$ ha$^{-1}$. The peak in stand biomass was caused by a high GPP of the pioneer species ($GPP_{pioneer} = 83$
$t_{ODM}$ ha$^{-1}$; Fig. S1.a). This defined the first phase for the years 0 to 40. After the ignition stage, the stand biomass fell slightly
until year 100 (Fig. 2) because of the rapidly declining pioneer biomass (stem exclusion phase), while the biomass of other
species increased. After 100 years, the stand biomass stabilised around 420 $t_{ODM}$ ha$^{-1}$a$^{-1}$ (average over years 100–300). The
steady state of the functional species composition was reached after 210 years (Fig. 2). In the last phase (gap dynamics), climax
species and species with intermediate light requirements fixed five times more carbon in biomass than pioneer species (GPP;
Fig. S1.a).





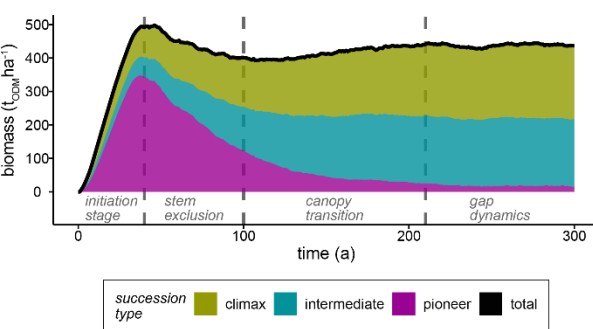

**Figure 2: The baseline scenario's aboveground biomass per successional type as fraction of total biomass for Paracou's *Terra Firma* forest in French Guiana. Dashed lines separate different phases of forest succession. (AGB: aboveground biomass, ODM: organic dry matter).**

Our simulation results reveal a sensitive response of biomass mortality to increased disturbance intensities (Fig. 3.a). At higher disturbance intensity, a higher biomass mortality with greater variance emerged. At the highest disturbance level, a peak in biomass mortality rate occurred around 0.12 a⁻¹ during the early phase of forest succession, before levelling off at a value of 0.08 a⁻¹ in the stable state (Fig. 3.a). Due to the higher biomass mortality, the light climate in the forest stand changed (Fig. 3.c). The pioneer species were able to establish quickly in gaps. Hence, the GPP of the pioneer species was highest among all species groups (Fig. S1.a), which also affected the productivity of the total stand (Fig. 3.d – 3.e). Please note that despite distinctly higher GPP in the case of higher rates of biomass mortality, NPP did not change for the different scenarios (Fig. 3.f). Thus, disturbance intensity had a strong influence on species composition (e.g. higher pioneer GPP; supplements Fig. S1.a), which led to lower values of LAI, biomass and mean forest height at the ecosystem level compared to the reference (Fig. 3.a – 3.c). In addition, from structural changes arose modified forest stand dynamics, with unique succession patterns depending on the intensity of disturbance.





**Figure 3: Development of (a) biomass mortality rate, (b) aboveground biomass, (c) leaf area index LAI, (d) forest height, (e) gross primary production GPP, and (f) net primary production NPP of *Terra Firma* forest stands for different disturbance intensities. Grey lines indicate the entire set of disturbance scenarios under varying basic mortality rates. (Biomass mortality: rate of biomass loss due to tree mortality, ODM: organic dry matter).**

Furthermore, we analysed how disturbance intensity affects the time needed to reach the equilibrium (Fig. 4.b). GPP responded particularly sensitive and inversely proportional, showing a strong decrease with rising disturbance levels. In contrast, other forest attributes, such as biomass and NPP, had altogether shorter equilibrium times than GPP, increasing directly proportional to the disturbance intensity.

Finally, we evaluated the effect of increasing tree mortality rates on the residence time of biomass (eq. 4) in forest stands taking forest succession into account (Fig. 4.c). The biomass residence time $\tau$ was halved at a five-time higher disturbance intensity compared to the baseline ($\tau_{(f=1)} = 34$ a, $sd_{(f=1)} = 13$ a; $\tau_{(f=5)} = 14$ a, $sd_{(f=5)} = 4$ a). Important forest properties are profoundly affected, if the functional species-composition, tree size distribution and dynamics of forests are changed.



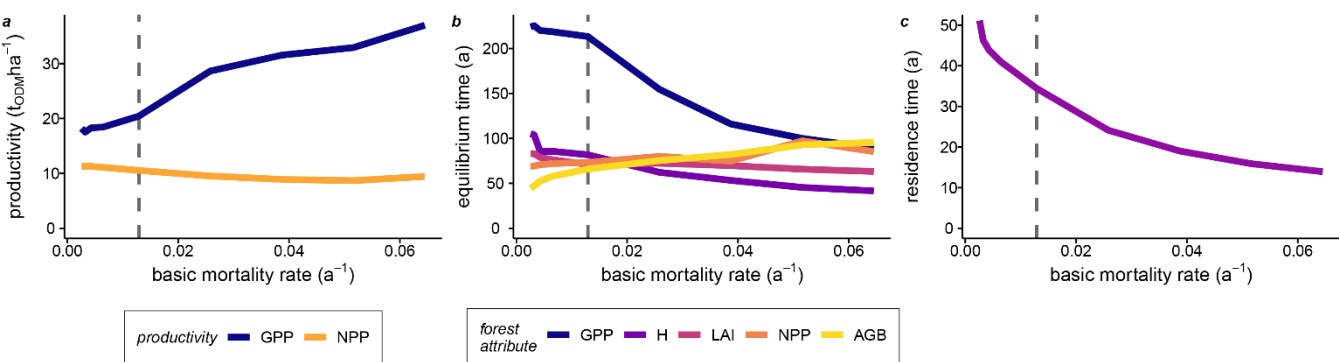

**Figure 4: Influence of disturbance intensity on Paracou's *Terra Firma* forests on (a) the mature forests' mean gross primary production and mean net primary production (averages over years 250 – 300), (b) the time until forest attributes reached the equilibrium and (c) the mean biomass residence times as reciprocal value of biomass mortality rates (cf. eq. 4; averages over years 0 – 300). Dashed lines indicate the baseline scenario. (GPP: gross primary production, NPP: net primary production, H: forest height, LAI: leaf area index, AGB: aboveground biomass, ODM: organic dry matter).**

### 3.2    Estimation of mortality rates from forest attributes

In a further analysis, we assessed how biomass mortality can be derived from different proxy variables, such as mean forest height, biomass, and leaf area index. We tested the relationships between several single forest attributes and biomass mortality, but did not find a simple relationship (Fig. 5). The relationships are strongly influenced by forest succession. Biomass mortality rates showed a widely scattered range of values and thus unclear relationships to all forest attributes during the initiation stage (Fig. 5). For instance, the LAI of less disturbed old-growth forests (i.e. LAI = 4 during gap dynamics stage) indicated similar biomass mortality rates as for forests in an early stage of succession.

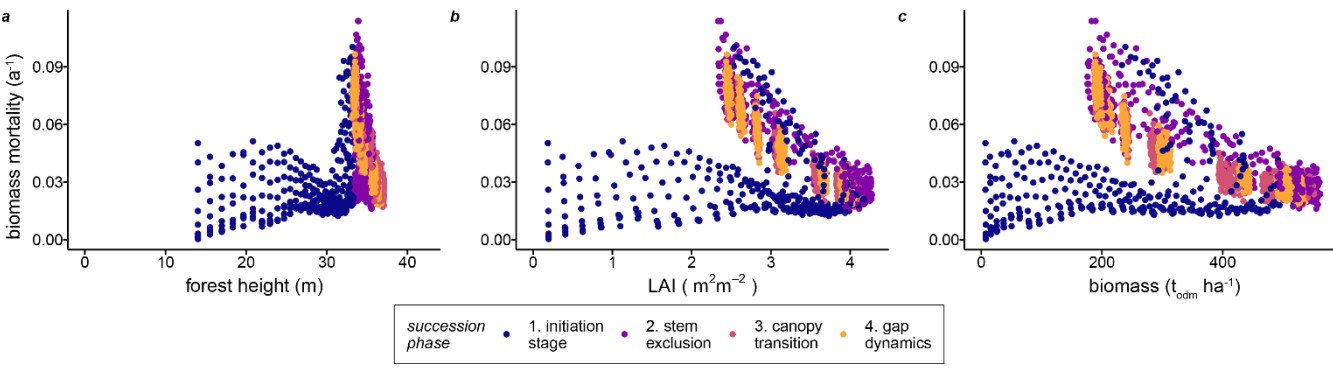

**Figure 5: Simulated biomass mortality as a function of (a) forest height, (b) leaf area index, and (c) aboveground biomass of *Terra Firma* forests in Paracou. Colours indicate the succession phases (cf. Fig. 2). (biomass: aboveground biomass, LAI: leaf area index, biomass mortality: rate of biomass loss due to tree mortality).**


When combined in a multivariate linear regression model, however, the training data of LAI and forest height explained the biomass mortality rates of forests quite well ($R^2 = 0.9484$, RMSE = 0.0106, p-value < 0.001; Fig. 6; Tab. 2, Fig. S2.a). The





obtained residuals were normally distributed around the expected value ($E$ ($m_{AGB}$) = 0.0; Fig. S2.b) and, depending on the fitted biomass mortality rates, they were homoscedastic with almost no trend (Fig. S2.c). The LAI influences negatively ($\beta_2$ = -

0.0341) and thus has the highest weight, followed by positively correlated forest height ($\beta_1$ = 0.0045). The obtained linear regression model for differently disturbed forests is given as follows:

$$m_{AGB} = 0.0045 \cdot H - 0.0341 \cdot L + \varepsilon \qquad (6).$$

GPP and NPP were not included in the multivariate linear model, because they did not improve the estimation of the biomass mortality rate.

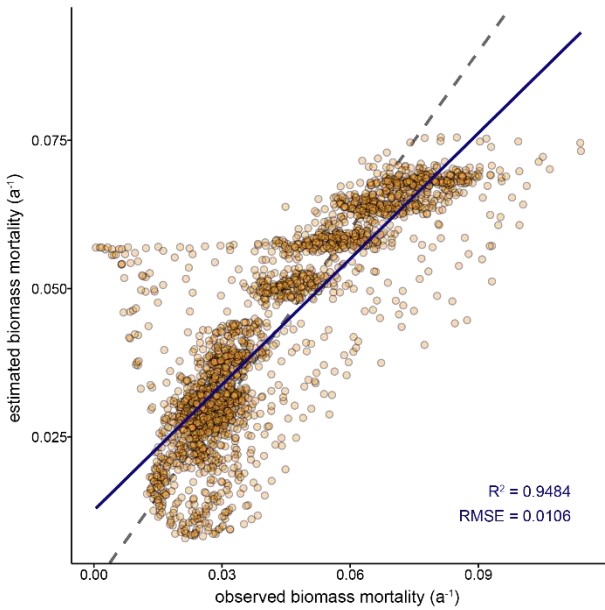


**Figure 6: Biomass mortality (rates of biomass loss due to tree mortality) estimated from forest height and LAI (equation 6) using the FORMIND model. The blue line indicates the mean deviation of the estimated biomass mortality rates from the simulated ones. The grey dashed line shows the 1:1-line. For more results of the multivariate linear regression model fit see Tab. 2.**

**Table 2: Results of multivariate linear regression model fit of mean forest height and LAI. The standard error is in parenthesis**
**and p-value of *** < 0.001 (H: mean forest height, L: LAI, β: coefficients).**

| Coefficients of explanatory variables | Estimated coefficients |
|---|---|
| $\beta_H$ | 0.0045*** (3.8e-05) |
| $\beta_L$ | -0.034076 *** (3.8e-04) |
| Data base: | |
| *adjusted R²* | 0.948402 |
| *RMSE* | 0.010609 |



### 3.3 Spatial distribution of biomass mortality

By combining simulation data with the maps of LAI and forest height from remote sensing (Myneni et al., 2015; Simard et al., 2011), a biomass mortality map with a resolution of roughly 2 km² was derived for French Guiana (Fig. 7). Based on this map,

we obtained a mean biomass mortality rate of 0.030 a⁻¹, (standard deviation of 0.014 a⁻¹); this corresponds to an average biomass residence time τ of 41 years (sd$_\tau$ = 19 a; cf. Fig. S4). The values of biomass mortality vary between regions with higher rates in the southern part and lower rates in the northern part of the country. The highest biomass mortality rates can be observed in the centre and at the south-western and eastern country borders (m$_{AGB}$ > 0.07). Such high values result from a combination of tall forest height together with low LAI (Fig. S5). In the region surrounding the Paracou study site, the biomass

mortality rate has a value of 0.020 a⁻¹, which corresponds well to the simulated mean biomass mortality under the baseline scenario (m$_{AGB,bl}$ = 0.031; sd$_{bl}$ = 0.007).

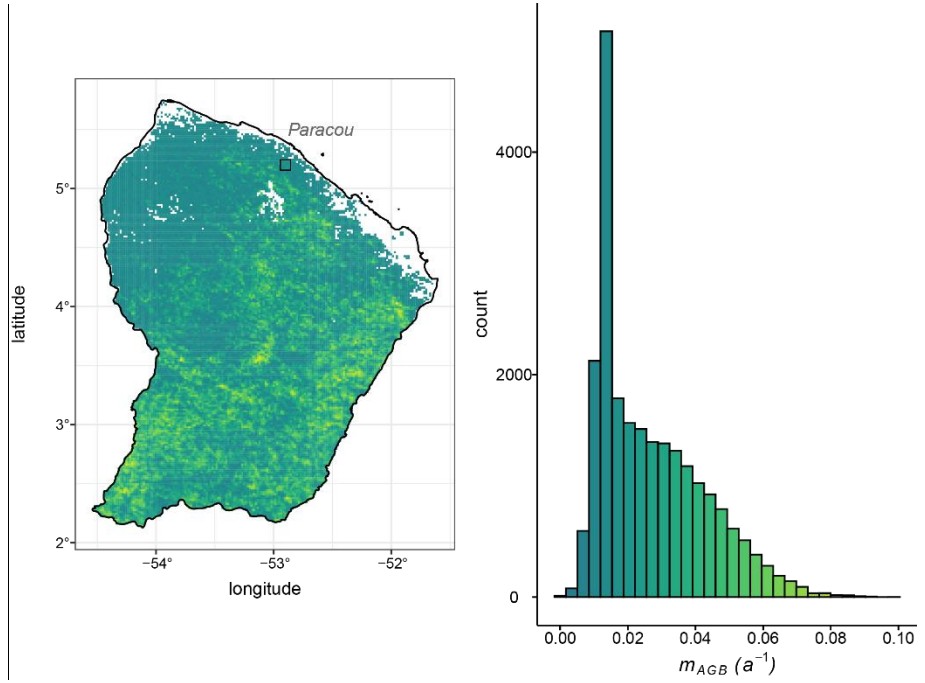

**Figure 7: (left) Map of biomass mortality distribution and (right) its histogram for forests of French Guiana at ~2 km resolution. To estimate these rates, remote sensing was combined with a forest model that simulated the forest succession. The black square**
**indicates the location of the Paracou site. Leaf area index and forest height were used as proxy variables for the underlying multivariate linear regression.**



## 4  Discussion

### 4.1    Mechanism of tropical forests in dealing with increasing intensity of tree mortality

In this study, we analysed succession dynamics in tropical forests in relation to tree mortality. It was possible to demonstrate
that most of the analysed forest attributes (biomass, forest height, LAI, GPP, biomass mortality) had a specific response during
succession. Moreover, we were able to show that biomass mortality rates are strongly affected by the succession dynamics, as
well as by the disturbance intensity. For each disturbance scenario, the period until the stand's equilibrium was reached differed
in duration. Also, the mean residence time of biomass, i.e., the reciprocal value of biomass mortality (eq. 4), varied
considerably. The reasons for the unique succession patterns of each forest attribute are multiple. Succession dynamics are
influenced by assimilation rates (e.g. photosynthesis rate, light requirement) and physiognomic characteristics (e.g. maximum
stem diameter increment rates, maximum height, and wood density), which are specific to each species group (Hiltner et al.,
2018). Functional traits are decisive in order to simulate the succession dynamics in forests, because they determine the
competitiveness of species groups (Fischer, et al., 2018; Rüger et al., 2019).

The relationship between succession and tree mortality was investigated in empirical studies to estimate the mortality in forests.
Aubry-Kientz et al. (2013) introduced a method, which estimated the tree mortality probability of *Terra Firma* forests at
Paracou. Similar to our results, they found that the tree mortality probability depends on successional stages of forests as well
as functional traits of species, such as specific leaf area, wood density, stem diameter increment, and potential height.

Interestingly, we observed nearly similar NPP values for different levels of tree mortality for forests in equilibrium. Erb et al.
(2016) argued that NPP of vegetation is more or less independent from disturbance intensity, which is confirmed by our results.
The observed stability of NPP under different disturbance regimes can be explained by shifts within the functional species
composition of the forest stands. The pioneer species, which typically have lower wood density and lower potential height
than slow-growing climax and intermediate species (Chave et al., 2009; Zanne et al., 2009) store less carbon in living biomass.
Since pioneer species grow faster, they can bind as much carbon per time as slow growing climax species. Therefore, similar
NPP values are obtained at stand level with higher disturbance intensity, although the individual trees show different growth
behaviour.

### 4.2    Performance of the regression model for estimating biomass mortality

One of the main findings of this study is that the biomass mortality rates of *Terra Firma* forests can be estimated using simple
relationships between the forest attributes. We selected forest attributes that provide information about forest structure and
productivity, and that showed distinct succession patterns. On a tree level, Aubry-Kientz et al. (2013) used functional traits,
such as potential tree height and specific leaf area, to estimate the probability of tree mortality. Based on large scale remote
sensing observations tree height was identified as the strongest predictor of tree mortality, with large trees having twice the





mortality rate of small ones, while environmental drivers controlling the intensity of the height-mortality relationship (Stovall et al., 2019). These results suggest using forest height and LAI as proxy variables to estimate the mortality of forest stands.

Despite the simplicity of the regression model, including only two explanatory variables, its statistical performance proved to
be robust (cf. eq. 6; Tab. 2, supplements Fig. S2). Thus, it was possible to derive biomass mortality from LAI and forest height for undisturbed and disturbed forests. It was important that the signs of the regression coefficients $\beta_i$ of our linear model plausibly reflected the observed relationships in the field. In the regression model, forest height is directly proportional and LAI indirectly proportional to the biomass mortality of the forest stands, e.g. tall forests with a low LAI result in higher biomass mortality rates (cf. Fig. S5).

Using a forest model for deriving relationships between different forest attributes has several advantages: first, the simulation data of LAI and forest height were generated mechanistically, integrating a broad spectrum of information about forest dynamics emerging from different physiological processes. This can lead to a lower level of noise in the simulation data compared to observed field data. Nevertheless, forest models also include stochastic processes, e.g. for basic tree mortality (Bugmann, 2001; Fischer et al., 2016; Shugart, 2002). By using plant functional types for the simulation of forest dynamics,
we reduced possible uncertainty in species traits. Simplifications allow a transferability of the used regression analysis to forests of similar succession dynamics and disturbance intensities as the simulated forests of Paracou. This also enabled the spatial extrapolation of biomass mortality rates to *Terra Firma* forests of whole French Guiana. With the approach pursued here, it is possible to derive regression models for estimating biomass mortality for other locations worldwide. It remains to be investigated whether LAI and forest height are also suitable for other forest types as explanatory variables of biomass
mortality rates.

### 4.3 Mapping of biomass mortality rates on a large scale

We combined remote sensing maps of forest height (Simard et al., 2011) and LAI (Myneni et al., 2015) with forest modelling for deriving maps of biomass mortality rates for French Guiana. Capabilities for improved projection are indispensable in the context of man-made climate and land use changes (IPCC, 2014, 2018). Remote sensing by airborne and satellite-based
instruments provides large-scale data on forests, such as forest height (Simard et al., 2011), as well as leaf area index (Myneni et al., 2015). However, remote sensing can measure only at certain time points, hence, the successional stage of forest variables is uncertain. Such forest dynamics can be simulated by individual- and process-based forest models. A combination of remote sensing data and forest models therefore has the potential to improve the prediction of large-scale ecosystem dynamics (Plummer, 2000; Shugart et al., 2015).

Forests can be in different successional stages due to disturbance, which influences forest height and LAI (Dubayah et al., 2010; Kim et al., 2017). In the forest height and LAI maps, disturbed regions can be detected, which have been identified as disturbed areas in other studies (Asner and Alencar, 2010; Piponiot et al., 2016a; Stach et al., 2009). For example, in flood plains of lakes and rivers, along the coast, near roads and settlements, or in secondary forests of French Guiana, where forest





height is on average lower than in primary forests (Piponiot et al., 2016a; Stach et al., 2009; forest height map after Simard et
al. (2011) in Fig. S 3.a). Additionally, the crown structure of tropical forests, represented by LAI, decreases as a result of water
deficit during drought (Asner and Alencar, 2010; Pfeifer et al., 2018; LAI map after Myneni et al. (2015) in Fig. S 3.b).

### 4.4 Introduction of an alternative method to estimate biomass residence time

Information on the carbon balance of forests is important to quantify the biomass accumulation rates in trees. Various studies
estimated the residence time of biomass, which we defined here as the reciprocal value of biomass mortality, in forests
worldwide (Carvalhais et al., 2014; Erb et al., 2016; Pugh et al., 2019). Carvalhais et al. (2014) were the first who estimated
biomass residence times for forests in equilibrium from biomass and GPP (cf. eq.3: $\tau = AGB_{total} \cdot GPP^{-1}$). For the French
Guiana region, they estimated biomass residence times of around 20–40 years and discussed that disturbances can shorten the
biomass residence time by increasing biomass mortality rates. Our study quantifies in how far disturbances lead to a higher
biomass mortality and thus to a shorter biomass residence time.

Erb et al. (2016) observed decreases of the biomass residence time caused by land use. They found residence times of 20–30
years for the French Guiana region, which is similar to our results (Fig. S4). Pugh et al. (2019) showed that stand-replacing
disturbances also affect the biomass residence times negatively, which means they become shorter. We found that biomass
residence time is strongly affected by succession dynamics and disturbance intensity. For French Guiana, we found a mean
biomass residence time of 41 years. We derived an alternative framework to estimate the residence time from biomass
mortality, which allows both of them to be modelled in a simple way considering succession dynamics and disturbances. This
method can be applied to forests in equilibrium, but also to forests in earlier stages of succession, which can emerge due to
disturbances and logging.

### 4.5 Outlook

Our simulation results revealed complex relationships between tree mortality and biomass mortality. The growth stage of a
tree evidently has an effect on tree mortality, which typically results in a U-shaped relationship of tree mortality as a function
of tree size distribution in forests (Aubry-Kientz et al., 2013; Muller-Landau et al., 2006). With regard to tree age, it is more
likely that the youngest and oldest trees will die (Aubry-Kientz et al., 2013; Rüger et al., 2011), e.g. due to intense competition
for light and space between the juvenile trees in the understory and senescence of the old trees in the canopy layer. Such
mortality processes are often represented in forest models (Bugmann et al., 2019). Although empirical mortality algorithms
describing main causes of tree mortality and their effects on entire ecosystems mechanistically (e.g. self-thinning, dying of
other trees by crushing, and growth dependent mortality) have already been developed, other causes of tree mortality with
unclear signals are often summarised as stochastic processes (Bugmann et al., 2019; Hülsmann et al., 2017, 2018). In our
study, biomass mortality at the stand level arose from different mortality processes at tree level (competition due to crowding,
dying of other trees by crushing, growth-dependency, gap formation, and basic mortality); but includes also altered tree



mortality due to climate change effects and other external drivers. Compared to the U-shaped tree mortality distribution, the biomass mortality rates of a forest stand depended in a more complex way from different forest attributes (e.g. LAI, forest height).

In our study, the effects of disturbance were represented in a simplified manner by modifying the basic mortality rate. In addition, we analysed disturbances that permanently increased the tree mortality rate in the forests. However, it is also needed

to analyse effects of discrete or continuously changing disturbance patterns. Impacts of single discrete disturbance events (i.e. selective logging) on the dynamics of *Terra Firma* forests were studied by Hiltner et al. (2018). A follow-up study is in preparation, where repeated logging events are investigated under continuously changing air temperature and precipitation.

It was also found that temporal patterns of regeneration can change after disturbances, e.g. due to modifications in seed mortality of specific tree species. Such changes influence competitive processes of trees within communities (Dantas de Paula

et al., 2018). Here we did not consider the influence of disturbance on regeneration processes. This should be considered in future studies.

With regard to up-scaling the biomass mortality rates, there are three important aspects: Firstly, it is important to verify the quality of the forest model parameterisation with field data, like it was done by Hiltner et al. (2018), who analysed biomass dynamics, tree size distribution, and functional species composition as well as compared model results with data from forest

inventories of the Paracou study site. Secondly, a regression model predicting biomass mortality rates is valid only for a certain type of forest. In up-scaling biomass mortality rates from stand level to landscape level, we assumed the pre-dominance of the same type of forest, here the *Terra Firma* forests in French Guiana (Hammond, 2005). For this forest type, Stach et al. (2009) calculated a forest cover of 95 % of the country's land area. Thirdly, site parameters across entire landscapes can be heterogeneous, affecting forest dynamics and structure. Various studies demonstrated that natural environmental factors such

as soil properties (Rödig et al., 2017), relief (Guitet et al., 2018), and climatic variations (Rödig et al., 2017; Wagner et al., 2012), but also logging history (Hiltner et al., 2018; Piponiot et al., 2016b, 2019) can affect the succession dynamics of *Terra Firma* forests. In further investigations, it is recommended to implement climatic or topographic parameters in order to further improve the approach developed here.

## 5   Conclusions

Here, we developed a framework for estimating biomass mortality in tropical forests. We analysed effects of tree mortality under different disturbance intensities and its relation to forest productivity, and biomass based on the example of *Terra Firma* forests of French Guiana. By quantifying such effects through simulation experiments, it was possible to derive a linear relationship between biomass mortality and other forest attributes. Our approach revealed a strong influence of succession dynamics or disturbance intensity on biomass mortality of forests.
We combined individual-based forest modelling with remote sensing, so that an upscaling from local forest stands to an entire landscape was enabled. The resulting map of biomass mortality rates indicates that more biomass is dying in the central, southern and eastern regions than in the northern parts of French Guiana. The forest areas in the north and north-east are more used for timber production, agricultural activities and housing (Bovolo et al., 2018; Stach et al., 2009), whereas the forest areas in the south are predominantly natural rainforests (Hammond, 2005).

The approach we developed here can be easily transferred to other forest biomes using forest models that capture biome-specific forest dynamics (e.g. for boreal and temperate forests) and available remote sensing products. Estimating the spatio-temporal distribution of forests' biomass mortality rates has recently been identified as particularly relevant for the monitoring of mortality hotspots (Hartmann et al., 2018). Moreover, an improved estimation of the residence times of carbon in forest stands is possible, so that uncertainties in the global carbon cycle (Friend et al., 2014) can be reduced.

**Competing Interest**

The authors declare no conflict of interest. The funders had no role in the design of the study, in the collection, analyses, or interpretation of data, in the writing of the manuscript, and in the decision to publish the results.

**Acknowledgements**

We would like to sincerely thank *Dr. Nuno Carvalhais* and *Prof. Dr. Achim Bräuning* for fruitful discussions of the simulation
results and the manuscript. U.H. would like to thank *A. Keberer* for his remarkable assistance. U.H. was funded by the *German Federal Environmental Foundation – DBU* [AZ 20015/398] and the programme '*Realization of Equal Opportunities for Women in Research and Teaching*' *– FFL* of Friedrich-Alexander-University Erlangen-Nuremberg.

**Author's contributions**

U.H., A.H., and R.F. conceived and designed the experiments; U.H. acquired and managed the data; U.H. performed the
simulations; U.H., A.H., and R.F. contributed in analysis and discussion; U.H. wrote the manuscript; U.H., A.H., and R.F. reviewed the manuscript.

**Data availability**

The data and code are available on request.





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
