# Peer review of "Importance of succession in estimating biomass loss: Combining remote sensing and individual-based forest models"

_Biogeosciences, 2020_

## Referee Comment (RC1) · Anonymous Referee #1 · 3 Sep 2020

Overview: The authors attempt to integrate simulation model predictions, remote sensing, and field data to scale above ground necromass in French Guiana. The model predictions of biomass mortality under differing mortality rates are then used to fit a statistical linear multiple regression model with the covariates of LAI and height. The statistical model is used to upscale biomass mortality across the entirety of French Guiana.

General Comments: It is clear the authors have done a lot of analysis on these model simulations. The figures are nice and clear, and the text is mostly well detailed. However I have major reservations about the underlying reasoning of the manuscript. The

model might work nicely - I don't really know and it doesn't appear anyone would if there was not a large body of field data to test it with. Moreover I don't see the utility of 'upscaling' to the entirety of FG from one plot. Sure you can generate an estimate, but is the estimate viable or defensible? There are certainly some interesting aspects to the simulations such as the interaction between PFT and disturbance frequency - yet that is not really what the outcome of the manuscript is focused upon. Overall, these issues and the following underlie my objections to the methodological approach used here, and subsequent conclusions derived.

1)Does the paper address relevant scientific questions within the scope of BG? I suppose this would fall within the scope.

2) Does the paper present novel concepts, ideas, tools, or data? This might be a novel application of FORMIND simulation results. I am not deeply read in the FORMIND literature. Otherwise, I think these approaches have largely been attempted before.

3) Are substantial conclusions reached? An estimate of mortality and biomass turnover for the entirety of French Guiana is derived.

4) Are the scientific methods and assumptions valid and clearly outlined? No, this is lacking with respect to the assumptions of the underlying model products used to make the assertions of the results.

5) Are the results sufficient to support the interpretations and conclusions? I do not believe so. They present a rate of mortality, but there is no ground based observation presented to compare it to, neither is an appropriate comparison made with field derived tree mortality estimates from the region.

6) Is the description of experiments and calculations sufficiently complete and precise to allow their reproduction by fellow scientists (traceability of results)? No, the specific version and parameterization of the FORMIND model and its outputs are not made available. The large scale predictions are also not made available.

7) Do the authors give proper credit to related work and clearly indicate their own new/original contribution? No, I believe they actually miss a lot of relevant work - especially with respect to field observations. It is possible because some of this work is contrary to their conclusions.

8) Does the title clearly reflect the contents of the paper? Not exactly. LAI is not entirely representative of successional state. Succession usually has a species assemblage connotation, which cannot be derived from the remote sensing products used here. There are many intrinsic edaphic and topographic effects that can also limit LAI, in addition to intra & inter-annual variability of LAI from phenological responses to anomalies of climate.

9) Does the abstract provide a concise and complete summary? The abstract is not concise, and I think some of the statements should need to be edited to make it completely clear that every result presented here is conditional upon the veracity of the predictions of FORMIND being simulated for the Paracou plot. Also mention of climate change is made, but that's not really at all thematic of the manuscript.

10) Is the overall presentation well structured and clear? I was confused by some aspects of the methods, but the structure of the presentation seems ok.

11) Is the language fluent and precise? There are some areas where the language is a bit informal, but this could be easily remedied and is not a major concern. Some sentences should be re-written in the 'direct voice'. The last sentence of the conclusion reads very awkwardly as is.

12) Are mathematical formulae, symbols, abbreviations, and units correctly defined and used? I think this is mostly ok. There might be a small issue here with terminology. For example, what is called "rate of biomass loss due to tree mortality" is actually a proportion.

13) Should any parts of the paper (text, formulae, figures, tables) be clarified, reduced,

combined, or eliminated? The text does seem a bit long. The figures look nice.

14) Are the number and quality of references appropriate? No, I think the references are very much inadequate. There is a large omission of comparison with field based studies, and tropical forest remote sensing studies.

15) Is the amount and quality of supplementary material appropriate? It seems ok. The model, simulation outputs, and derived country level predictions should be made available (without requesting access).

RS LAI data: Changes in LAI are not at all indicative of tree mortality - especially at the relatively coarse 500m MODIS scale. Intrinsic biological phenology and drought responses can also trigger large fluctuations in LAI. I think something on the scale of a large windthrow event would be required to really reduce the LAI at the scale of resolution in the MODIS product. I see this as a problem that undermines the underlying approximation of mortality for the manuscript - and by extension I think undermines the effort of upscaling mortality. Lines 60:63 also seem to make this point.

Statistical model: The high R2 of the statistical model approximating mortality derived from the simulation model is not very meaningful when (1) it is completely unclear that the model can accurately simulate mortality. (2) There also appears to be a scale mismatch between the simulation outputs the statistical model was fit with, and the RS derived inputs used for upscaling it. Was the model fit with 40x40 m subsets or 60 ha? It is not really clear. Given the number of points in figure 6, I am guessing it is the 40x40m subsets. The native scale of the tree height product (which is also a model derived product) is 100 ha. I am sceptical of fitting a model on simulated mortality predictions of 0.16 ha, and then applying it at 100 ha.

Numerous papers have shown that tree mortality and necromass do not scale linearly. If this manuscript was actually based upon field data (which it is not), then perhaps there would be merit to this counter argument. However, the results of this manuscript and its thesis is effectively entirely based upon simulations. The authors do not seem

to acknowledge previous research on the topic - which again, is strongly contrary to the results presented here.

I think it is extremely speculative to assume that changes in a modeled LAI estimate are proportional to % mortality, or total necromass. Virtually all allometric equations for biomass are nonlinear. A lot of hard work has been done in this area. See Marra et al. 2016 Biogeosciences. A number of papers have shown that non-linear size responses occur with common drivers of tree mortality. Droughts are thought to disproportionately kill large trees (Nepstad et al 2007 Ecology). The same goes for wind mortality (Rifai et al 2016 Ecological Applications), fire & wind (Silvério et al 2018 Journal of Ecology). But otherwise there are so many other drivers of mortality that cannot simply be linearly scaled by height and LAI.

Barlow et al 2003 Ecology Letters; Fauset et al., 2019 Frontiers in Earth Science; Fisher et al., 2009 Ecology Letters; Chambers et al., 2009 Ecology Letters; Chambers et al 2013 PNAS; Marra et al 2014 PlosOne; Marra et al., 2018 Global Change Biology; McDowell et al 2018 New Phytologist; Negrón-Juarez et al., 2018; Negrón-Juarez et al., 2010 Geophysical Research Letters; Rifai et al., 2016 Ecological Applications; Sivério et al 2019 Journal of Ecology; and many many more.

The x & y axes on figure 6 should be flipped in my opinion. There are some countering opinions on this, but typically observations are on the y-axis. However, there appears to be some non-linear influence between the quasi-observations and simulations of biomass mortality (necromass) that is not (or perhaps cannot be) captured with the linear regression.

Scaling from one forest plot to a large region: Again it should be made absolutely clear, repeatedly and throughout the abstract that these results are based on model simulations. This includes the estimates of LAI, which are indeed modeled and not directly observed. The arguments of this manuscript seems to be heavily dependent upon the veracity of the MCD15 product - however any optical remote sensing product

has saturation effects when forest canopies are dense with an LAI > 4. The assumption of the Paracou forest plot being representative of the entirety of French Guiana is exceptionally misplaced. The climate of Paracou is influenced by its proximity to the Atlantic. The supplemental figure S3, for example shows two aspects of how this site cannot represent the entirety of French Guiana. No one site can really be claimed to be representative of such a large area. The simulations do not appear to be very realistic. Paracou exists upon relatively infertile soil with extremely limited Phosphorus. The simulation of approaching 500 Mg Biomass/Ha in less than 50 years is inconceivable with field measurements of NPP. These numbers should be compared with field observations in around the tropical forests of the Guiana shield.

Data availability: The value of this model focused manuscript is markedly reduced if the data and code are not openly available. I think the unavailability of the data and code is also contrary to the journal guidelines (https://www.biogeosciences.net/about/data_policy.html). If the data is available, then make it available - otherwise a detailed statement is required as to why it is not available. The need to contact the authors is especially burdensome upon the reader, and is unlikely to be robust against the effects of time. Can the authors really guarantee they will always be around to provide the data and code when requested? Finally, even if FORMIND is available through other means, the results of this manuscript are unreproducible if it is not specific to the exact variant of FORMIND used in this manuscript.

Line Comments: Figure 1 caption: What is meant by rejuvenation?

Figure 3 caption: Put parentheses around acronyms that are being defined.

Figure 7: Is the histogram meant to serve as a colorbar for the left panel? This is not very clear if so. It would be better to add a color bar indicating what the color gradient signifies.

Table 2: Report the intercept value of the linear regression.

[Figure]

20: Multivariate regression is when there are multiple response variables in the same regression model. Perhaps 'multiple linear regression' is meant?

25: I cannot tell if this is in reference to a model simulation or field observations?

38: The Pan 2011 estimate of 2.8 is on the higher end and was assembled more or less haphazardly from the available forest census data and country level reports. More recent estimates are available.

41: You might see Korner 2003 Science, Chambers et al 2013 PNAS, and Fisher et al 2008 Ecology Letters.

67-68: I don't see how this statement can be justified.

81-83: I find this hard to justify. See comparisons on Paracou and Nouragues.

86-87: Competition for water is a major axis not mentioned. The Guiana shield has been struck numerous times by severe drought effects.

95: About the regression model, what is the response and what is the covariate?

105-110: The paper using FORMIND v3.2 appears to be focused upon estimation of biomass with respect to changes in forest management. The parameterized version of the model does not appear to be available from that publication either.

119-120: I find the model simulating mortality with arbitrarily distributed spatial patterns to be extremely implausible. Wind, fire, floods ∼ these all have a distinct spatial component. This spatial component has implications for who dies, and the post-disturbance light environment. Moreover, disturbance in reality is a punctuated event. If I read the section 2.2.2 correctly - the imposed disturbance intensity is actually just a multiplier on the baseline mortality rate. I don't think this is really anywhere near representative of disturbance in tropical forests.

160: This does not make sense to me. Allometric equations for biomass are typically nonlinear. See the widely used models including height in Chave et al., 2014 Global

Change Biology.

175-180: I don't understand what exactly was done here. Tab S1 in the supplement is actually a paragraph. Eq 5 and 6 appear to be the same equation. Was mortality derived from the simulation model? If this was the case, I don't think there is anything that can really justify this. The manuscript appears to be about upscaling mortality with remote sensing data - but the core critical part, the mortality - is derived from a simulation model. This is making a very large number of assumptions, which I find implausible.

259: Why 2km2 when the coarsest RS data was 1km2?

340-345: I think comparison with field based estimates of tau is important. I suggest reading more into the actual forest census based literature to come up with more comparisons. The Erb 2016 & Carvalhais 2014 papers are focused upon simulation results, and I don't agree that 20-30 years is similar to 40 years.

390: Terra Firme (not Terra Firma) is more commonly used to refer to this type of tropical forest

390: 'successional' -> 'succession'

---

## Referee Comment (RC2) · Thomas Pugh (Referee) · 24 Nov 2020

This interesting study uses a forest model to explore how biomass mortality rate varies as a function of changes in stem mortality rate for a tropical forest location in French Guiana. It then uses these simulations to create a simple emulator linking indicators of successional stage and the resulting rate of biomass mortality. The emulator is then used to estimate biomass mortality rates across the whole of the country. The model has been previously been evaluated at the reference site used in this region. Overall, I find this a novel approach to investigate spatial variations in biomass mortality rates as a result of differences in forest age. Such efforts are important to provide baseline

levels of mortality against which future changes can be compared, as well as to provide insights into the mechanisms driving mortality rates and any associated trends. I would like to see the manuscript published, but prior to that there are several aspects that I think should be clarified or expanded upon, including some additional analysis to identify the robustness of the results.

Main comments

1. A stand-level relationship between height, LAI and biomass mortality rate, is being used to scale up across a broad geographical region. Forest height here is almost purely an indicator of age of the largest trees, since there is relatively little difference between the disturbance scenarios at equilibrium. LAI appears to be both an indicator of age and composition. Between them they appear to characterise well how biomass mortality changes over the successional sequence. But when using this relationship to scale up, what happens if resource availability is not constant over the region being scaled over? Different levels of resource availability may also influence height and LAI - I should imagine particularly in the equilibrium stage for height and LAI from ca. 80 years onwards. Does the derived relationship in Fig. 6 hold across a productivity gradient? This is touched on in the limitations discussed in section 4.5, but I think it really needs to be tested (and presumably would be relatively straightforward to do). Even if the productivity gradient across French Guiana is small (as effectively assumed on L103), I think it is important for readers to know how robust the relationship and method are for application to more diverse regions.

2. In a similar vein, is it appropriate to liken the increase in biomass mortality rate with forest height in this study (driven by a uniform mortality rate change) to the increase in stem mortality rate with individual height in Stovall et al. (2019) (L306)? I think the mechanisms are quite different. Biomass mortality rate would be expected to increase as forests approach equilibrium biomass, as the size of biomass losses must start to approach that of biomass growth. But this does not have to imply that stem mortality rates increase with tree size - it could simply be that the trees that are dying are typically

larger. This is distinct from a mechanism in which individual tree mortality rates scale with individual height (e.g. Holzwarth et al., 2013; Rowland et al., 2015; Stovall et al., 2019). I suggest to add a bit more nuance in the discussion of this point. As an alternative comparison, in section 4.2, can the regression slopes instead be linked to the biogeographical patterns for the wider region from Johnson et al. (2016)? These patterns have been linked to a gradient in disturbance intensity and whilst Johnson et al. present biomass and stem mortality, rather than height and biomass mortality, FORMIND is simulating all components, facilitating a comparison.

3. It would also be good to see some independent evaluation of the extrapolation. Whilst observations for biomass mortality in the region are likely rather hard to come by, how similar is FORMIND simulated height and LAI to the Simard et al. and MODIS data used for the extrapolation? Are they very close to each other, or is a correction factor needed to account for biases in one or the other? I wonder if you could also compare biomass mortality rate with that from other plots in the Guiana Shield provided in Brienen et al. (2015)?

4. In section 4.4 it is stated that the new framework allows to assess residence time as a functional of successional stage, but I think this is a bit misleading. The term residence time comes loaded with implications about how long carbon stays in the system. But in a transient system (as opposed to an equilibrium one), this does not hold for the kind of calculation used here, and during succession the deviation from equilibrium is quite marked. The mean time a molecule of C entering a 50-year-old forest remains in that forest will likely be very different to the reciprocal of the biomass mortality rate at 50 years, because that molecule is more likely than its predecessors to be incorporated into a longer-lived later-successional PFT. In comparison, the biomass mortality rate is equivalently normalised to biomass but does not come with the same implications - it is unambiguously the rate at which carbon is currently leaving the system at that moment. I suggest only to use the concept of residence time here when averaging over the whole region (and then to term it turnover time, following Sierra et al., 2017).

5. The result that net primary production remained stable is very interesting and neat. But can you add a bit more discussion about what this result is ultimately based on? To what extent is it an emergent outcome of the model, versus an assumption that went into the PFT parameterisation?

6. Equation 3 implies that GPP = AGB_dead, which cannot be the case, as autotrophic respiration, allocation to soft tissue and allocation belowground need to be subtracted from GPP in order to get to woody NPP (i.e. woody biomass increment), which would be considered equivalent to AGB_dead at equilibrium (assuming that AGB is only counting the woody component of the total biomass). So, the tau obtained from AGB_total/GPP would be much smaller than that from AGB_total/AGB_dead. As FORMIND simulates GPP, a turnover time metric for comparison with Carvalhais et al. (2014) could be calculated, but it should be defined separately to the biomass turnover time with respect to mortality.

7. The LAI and height products used for extrapolation have errors associated with them. To what extent do these errors propagate through to uncertainty in the biomass mortality rates? I think Fig. 7 should be associated with an error field at least based on the input uncertainty, if not also the uncertainty in the regression fit.

8. Why only 1 month of LAI data (L170)? Doesn't this expose your results to potential seasonal LAI variations? Wouldn't an average over several years provide a more like-for-like comparison with the model output?

Minor comments

line 33. Imprecise statement. 471 Pg C is much less than half of the terrestrial carbon stock (assuming vegetation + soils) given most estimates. Given the reference to Pan et al. (2011), I think you mean "forest carbon stocks"?

l47. What is an, "increase in associated physiological mechanisms"?

l51. I always find disturbance a slippery term which can be used to refer to a very wide

range of things. In this paragraph you give a general list of things that can be referred to as disturbances. I think you are defining disturbance as everything which is not related to competition, which is fine of course, but could you give an explicit definition of what is considered as disturbance for the purposes of this study?

L191. GPP is not the right indicator for a statement about "fixing five times more carbon in biomass" (see above). Can you show woody NPP? Or simply say, "fixing five times more carbon"?

Fig. 4. ODM is not included in the figure, just the caption.

L301. The statement needs refining. Tree height is the strongest predictor of tree mortality out of which basket of indicators? At the individual tree level other predictors can be very important (see e.g. Esquivel-Muelbert et al., 2020), so it needs to be clear what is being compared to what.

L366. I would suggest that biomass mortality rates depended on functional composition and level of divergence of C input and output fluxes from equilibrium, with LAI and forest height being indicators of these, not the drivers themselves.

L396. "more biomass is dying", or, "biomass is dying at a faster rate"? (because the map shows rates, rather than fluxes).

References

Brienen, R. J. W., Phillips, O. L., Feldpausch, T. R., Gloor, E., Baker, T. R., Lloyd, J., Lopez-Gonzalez, G., Monteagudo Mendoza, A., Malhi, Y., Lewis, S. L., Vásquez Martinez, R., Alexiades, M., Álvarez Dávila, E. A., Alvarez-Loayza, P., Andrade, A., Aragão, L. E. O. C., Araujo Murakami, A., Arets, E. J. M. M., Arroyo, L., Aymard C., G. A., Bánki, O. S., Baraloto, C., Barroso, J. G., Bonal, D., Boot, R., Camargo, J. L., Castilho, C., Chama, V., Chao, K.-J., Chave, J., Comiskey, J. A., Cornejo, F., Da Costa, L., De Oliveira, E. A., Di Fiore, A., Erwin, T. L., Fauset, S., Forsthofer, M., Grahame, S. E., Groot, N. E., Herault, B., Higuchi, N., Honorio C., E., Keeling, H.,

Killeen, T., Laurance, W., Laurance, S., Licona, J.-C., Magnussen, W. E., Marimon, B. S., Marimon-Junior, B. H., Mendoza, C., Neill, D., Nogueira, E. M., Nunez, P., Pallqui Camacho, N. C., Parada, A., Pardo, G., Peacock, J., Pena-Claros, M., Pickavance, G. C., Pitman, N. C. A., Poorter, L., Prieto, A., Quesada, C. A., Ramírez, F., Ramírez-Angulo, H., Restrepo, Z., Roopsind, A., Rudas, A., Salomão, R. P., Schwarz, M., Silva, N., Silva-Espejo, J. E., Silveira, M., Stropp, J., Talbot, J., Ter Steege, H., Teran-Aguilar, J., Terborgh, J., Thomas-Caesar, R., Toledo, M., Torello-Raventos, M., Umetsu, R. K., Van Der Heijden, G. M. F., Van Der Hout, P., Guimaraes Vieira, I., Vieira, S. A., Vilanova, E., Vos, V. and Zagt, R. J.: Long-term decline of the Amazon carbon sink, Nature, 519(7543), 344–348, doi:10.1038/nature14283, 2015.

Esquivel-Muelbert, A., Phillips, O. L., Brienen, R. J. W., Fauset, S., Sullivan, M. J. P., Baker, T. R., Chao, K. J., Feldpausch, T. R., Gloor, E., Higuchi, N., Houwing-Duistermaat, J., Lloyd, J., Liu, H., Malhi, Y., Marimon, B., Marimon Junior, B. H., Monteagudo-Mendoza, A., Poorter, L., Silveira, M., Torre, E. V., Dávila, E. A., del Aguila Pasquel, J., Almeida, E., Loayza, P. A., Andrade, A., Aragão, L. E. O. C., Araujo-Murakami, A., Arets, E., Arroyo, L., Aymard C, G. A., Baisie, M., Baraloto, C., Camargo, P. B., Barroso, J., Blanc, L., Bonal, D., Bongers, F., Boot, R., Brown, F., Burban, B., Camargo, J. L., Castro, W., Moscoso, V. C., Chave, J., Comiskey, J., Valverde, F. C., da Costa, A. L., Cardozo, N. D., Di Fiore, A., Dourdain, A., Erwin, T., Llampazo, G. F., Vieira, I. C. G., Herrera, R., Honorio Coronado, E., Huamantupa-Chuquimaco, I., Jimenez-Rojas, E., Killeen, T., Laurance, S., Laurance, W., Levesley, A., Lewis, S. L., Ladvocat, K. L. L. M., Lopez-Gonzalez, G., Lovejoy, T., Meir, P., Mendoza, C., Morandi, P., Neill, D., Nogueira Lima, A. J., Vargas, P. N., de Oliveira, E. A., Camacho, N. P., Pardo, G., Peacock, J., Peña-Claros, M., Peñuela-Mora, M. C., Pickavance, G., Pipoly, J., Pitman, N., Prieto, A., Pugh, T. A. M., Quesada, C., Ramirez-Angulo, H., de Almeida Reis, S. M., Rejou-Machain, M., Correa, Z. R., Bayona, L. R., Rudas, A., Salomão, R., Serrano, J., Espejo, J. S., Silva, N., Singh, J., Stahl, C., Stropp, J., Swamy, V., Talbot, J., ter Steege, H., et al.: Tree mode of death and mortality risk factors across Amazon forests, Nat. Commun., 11(1), doi:10.1038/s41467-020-18996-3, 2020.

Holzwarth, F., Kahl, A., Bauhus, J. and Wirth, C.: Many ways to die – partitioning tree mortality dynamics in a near-natural mixed deciduous forest, J. Ecol., 101, 220–230, doi:10.1111/1365-2745.12015, 2013.

Johnson, M. O., Galbraith, D., Gloor, E., H, D. D., Guimberteau, M., Rammig, A., Thonicke, K., Verbeeck, H., Monteagudo, A., Phillips, O. L., Brienen, R. J. W., Feldpausch, T. R., G, L. G., Fauset, S., Quesada, C. A., Christoffersen, B., Ciais, P., Gilvan, S., Kruijt, B., Meir, P., Moorcroft, P., Zhang, K., Alvarez, E. A., Amaral, I., Andrade, A., Aragao, L. E. O. C., Arets, E. J. M. M., Arroyo, L., Aymard, G. A., Baraloto, C., Barroso, J., Bonal, D., Boot, R., Camargo, J., Chave, J., F, C. V., Ferreira, L., Higuchi, N. and Honorio, E.: Variation in stem mortality rates determines patterns of aboveground biomass in Amazonian forests: implications for dynamic global vegetation models, Glob. Chang. Biol., 22(12), 3996–4013, doi:10.1111/gcb.13315, 2016.

Pan, Y., Birdsey, R. a, Fang, J., Houghton, R., Kauppi, P. E., Kurz, W. a, Phillips, O. L., Shvidenko, A., Lewis, S. L., Canadell, J. G., Ciais, P., Jackson, R. B., Pacala, S. W., McGuire, a D., Piao, S., Rautiainen, A., Sitch, S. and Hayes, D.: A large and persistent carbon sink in the world's forests., Science (80-. )., 333(6045), 988–93, doi:10.1126/science.1201609, 2011.

Rowland, L., da Costa, A. C. L., Galbraith, D. R., Oliveira, R. S., Binks, O. J., Oliveira, A. A. R., Pullen, A. M., Doughty, C. E., Metcalfe, D. B., Vasconcelos, S. S., Ferreira, L. V, Malhi, Y., Grace, J., Mencuccini, M. and Meir, P.: Death from drought in tropical forests is triggered by hydraulics not carbon starvation., Nature, 528(7580), 119–122, doi:10.1038/nature15539, 2015.

Sierra, C. A., Müller, M., Metzler, H., Manzoni, S. and Trumbore, S. E.: The muddle of ages, turnover, transit, and residence times in the carbon cycle, Glob. Chang. Biol., 23(5), 1763–1773, doi:10.1111/gcb.13556, 2017.

Stovall, A. E. L., Shugart, H. and Yang, X.: Tree height explains mortality risk during an intense drought, Nat. Commun., 10, 4385, doi:10.1038/s41467-019-12380-6, 2019.

---

## Author Comment (AC1) · 15 Dec 2020

**Reply on the revision 1 of the Manuscript BG-2020-264**

**Reviewer 1**

Overview
The authors attempt to integrate simulation model predictions, remote sensing, and field data to scale above ground necromass in French Guiana. The model predictions of biomass mortality under differing mortality rates are then used to fit a statistical linear multiple regression model with the covariates of LAI and height. The statistical model is used to upscale biomass mortality across the entirety of French Guiana.

General Comments
It is clear the authors have done a lot of analysis on these model simulations. The figures are nice and clear, and the text is mostly well detailed. However, I have major reservations about the underlying reasoning of the manuscript. The model might work nicely - I don't really know and it doesn't appear anyone would if there was not a large body of field data to test it with. Moreover, I don't see the utility of 'upscaling' to the entirety of FG from one plot. Sure, you can generate an estimate, but is the estimate viable or defensible? There are certainly some interesting aspects to the simulations such as the interaction between PFT and disturbance frequency – yet that is not really what the outcome of the manuscript is focused upon. Overall, these issues and the following underlie my objections to the methodological approach used here, and subsequent conclusions derived.

> Thank you very much for your helpful comments. Below you will find our replies to your comments (highlighted in blue). Based on your comments, we would incorporate the following main changes to the manuscript:
> 1. Comparing the estimation of biomass mortality from model simulations with empirical data;
> 2. Adding more information on empirical studies about tree mortality and biomass loss;
> 3. Revising the discussion on limitations of used remote sensing products (such as LAI, forest height);
> 4. Improving the text for upscaling from stand-scale to country-scale.

1) Does the paper address relevant scientific questions within the scope of BG? I suppose this would fall within the scope.

> Reply 1: Thank you.

2) Does the paper present novel concepts, ideas, tools, or data? This might be a novel application of FORMIND simulation results. I am not deeply read in the FORMIND literature. Otherwise, I think these approaches have largely been attempted before.

> Reply 2: Our study presents an innovative approach that combines dynamic forest modeling with remote sensing data and a statistical modeling approach. To our knowledge, such approach has not yet been used in the field of Forest Ecology.

3) Are substantial conclusions reached? An estimate of mortality and biomass turnover for the entirety of French Guiana is derived.

> Reply 3: Thank you.

4) Are the scientific methods and assumptions valid and clearly outlined? No, this is lacking with respect to the assumptions of the underlying model products used to make the assertions of the results.

Reply 4: Previous studies have shown that the forest model can reproduce typical properties of old-growth Terra Firme forests (e.g., biomass, stem size distribution, and functional species composition) as well as succession following disturbance (see Hiltner et al., 2018). For the baseline scenario, the mortality rates were calculated from forest inventories. Therefore, mortality would have to be well represented by the forest model.

In order to demonstrate that the forest model can reproduce typical biomass mortality values, we will compare our results with empirical data and determine the quality of the simulations. Furthermore, we will compare the estimations of our country-wide biomass mortality map with empirical data. To do so, we will use data thar are available from publications on French Guiana (e.g. Brienen et al., 2015; Soong et al., 2020).

5) Are the results enough to support the interpretations and conclusions? I do not believe so. They present a rate of mortality, but there is no ground-based observation presented to compare it to, neither is an appropriate comparison made with field derived tree mortality estimates from the region.

Reply 5: Thank you. For the reply to this comment, please see our reply 4.

6) Is the description of experiments and calculations sufficiently complete and precise to allow their reproduction by fellow scientists (traceability of results)? No, the specific version and parameterization of the FORMIND model and its outputs are not made available. The large-scale predictions are also not made available.

Reply 6: Thank you. The FORMIND parameterization (Hiltner et al., 2018) and FORMIND's source code can already be freely downloaded on www.formind.org. We will revise the text to clearly indicate it.

We will make the analyses files and the biomass mortality map of French Guiana freely available as online attachment.

The data from the MCD15A2H Version 6 Moderate Resolution Imaging Spectroradiometer (MODIS) of LAI and the forest height map can be downloaded for free (Myneni et al., 2015; Simard et al., 2011). We will clearly indicate the download source in the methods part (in Chap. 2.4.1).

7) Do the authors give proper credit to related work and clearly indicate their own new/original contribution? No, I believe they actually miss a lot of relevant work – especially with respect to field observations. It is possible because some of this work is contrary to their conclusions.

Reply 7: We appreciate the advice and the mentioned empirical studies. They further point out important aspects of the topic. We will take them into consideration during the revision of the manuscript.

We would have to point out that our assumptions for disturbances differ from those empirical studies, because we are dealing with scenarios in which the tree mortality is at a higher/lower level over the entire simulation period (> 100 years). This is different to an increase in tree mortality due to a disturbance event over a short period (e.g., intra-annual). We will make this difference clearer throughout the manuscript. The investigation of short-term disturbances would be interesting in perspective, which will be discussed in the manuscript.

8) Does the title clearly reflect the contents of the paper? Not exactly. LAI is not entirely representative of successional state. Succession usually has a species assemblage connotation, which cannot be derived from the remote sensing products used here. There are many intrinsic edaphic and topographic effects that can also limit LAI, in addition to intra & inter-annual variability of LAI from phenological responses to anomalies of climate.

> Reply 8: We agree with the reviewer, LAI alone is not a proxy for successional state. Therefore, we analyzed the forest succession that emerges from model-inherent processes accounting for species diversity and for interactions between trees. Only when forest height is added to the analysis, we got a good proxy for the successional state. We will present this fact more clearly in the results and will discuss the limitations of LAI.

9) Does the abstract provide a concise and complete summary? The abstract is not concise, and I think some of the statements should need to be edited to make it completely clear that every result presented here is conditional upon the veracity of the predictions of FORMIND being simulated for the Paracou plot. Also mention of climate change is made, but that's not really at all thematic of the manuscript.

> Reply 9: Thank you for the suggestion. We will revise the abstract and will remove the mention of climate change (e.g. lines 14-15).
> We will clarify that this study's results represent simulation experiments on the dynamics of Terra Firme forests in French Guiana.

10) Is the overall presentation well-structured and clear? I was confused by some aspects of the methods, but the structure of the presentation seems ok.

> Reply 10: Thank you. We will check the method part.

11) Is the language fluent and precise? There are some areas where the language is a bit informal, but this could be easily remedied and is not a major concern. Some sentences should be re-written in the 'direct voice'. The last sentence of the conclusion reads very awkwardly as is.

> Reply 11: Thank you for pointing this out. We will revise the wordings of the manuscript and use more the direct voice.

12) Are mathematical formulae, symbols, abbreviations, and units correctly defined and used? I think this is mostly ok. There might be a small issue here with terminology. For example, what is called "rate of biomass loss due to tree mortality" is actually a proportion.

> Reply 12: Thank you. We will change the formulation.

13) Should any parts of the paper (text, formulae, figures, tables) be clarified, reduced, combined, or eliminated? The text does seem a bit long. The figures look nice.

> Reply 13: Thank you. We will pay attention to a compact formulation during the revision.

14) Are the number and quality of references appropriate? No, I think the references are very much inadequate. There is a large omission of comparison with field-based studies, and tropical forest remote sensing studies.

> Reply 14: We will add references on empirical studies and remote sensing, please see our reply 7.

15) Is the amount and quality of supplementary material appropriate? It seems ok. The model, simulation outputs, and derived country level predictions should be made available (without requesting access).

Reply 15: Thank you. For the reply to this comment, please see our reply 6.

16) RS LAI data: Changes in LAI are not at all indicative of tree mortality - especially at the relatively coarse 500m MODIS scale. Intrinsic biological phenology and drought responses can also trigger large fluctuations in LAI. I think something on the scale of a large windthrow event would be required to really reduce the LAI at the scale of resolution in the MODIS product. I see this as a problem that undermines the underlying approximation of mortality for the manuscript - and by extension I think undermines the effort of upscaling mortality. Lines 60:63 also seem to make this point.

Reply 16: We agree. As the reviewer already describes, the MODIS LAI product is limited to large-scale changes in LAI (e.g., after droughts, wind-throw, etc.). We will add the limitations of LAI in the discussion. To better evaluate the impact of potential uncertainties in the LAI product on our results, we intend to perform a sensitivity analysis of the LAI. The aim is to evaluate the LAI uncertainty on our results.

17) Statistical model: 1) The high R2 of the statistical model approximating mortality derived from the simulation model is not very meaningful when it is completely unclear that the model can accurately simulate mortality. 2) There also appears to be a scale mismatch between the simulation outputs the statistical model was fit with, and the RS derived inputs used for upscaling it. Was the model fit with 40x40 m subsets or 60 ha? It is not really clear. Given the number of points in figure 6, I am guessing it is the 40x40m subsets. The native scale of the tree height product (which is also a model derived product) is 100 ha. I am skeptical of fitting a model on simulated mortality predictions of 0.16 ha, and then applying it at 100 ha.

Reply 17: Thank you.
@ 1) For our more detailed response regarding the quality of the forest model, please see our reply 4.
@ 2) The forest model simulates 16 ha forest stands (400 m x 400 m) with a spatial resolution of 20 m x 20 m. This fine resolution allows us to scale up to the scale of remote sensing products. We used a resolution of 2 km for the biomass mortality map, although the input data are available in 0.5 km (LAI, Myneni et al., 2015) and 1 km (forest height, Simard et al., 2011). A coarser scale results in fewer uncertainties being reflected in the map (Fig. 7). We will present these facts more clearly. We will also better explain the spatial resolution of the data and simulation results.

18) Numerous papers have shown that tree mortality and necromass do not scale linearly. If this manuscript was actually based upon field data (which it is not), then perhaps there would be merit to this counter argument. However, the results of this manuscript and its thesis is effectively entirely based upon simulations. The authors do not seem to acknowledge previous research on the topic - which again, is strongly contrary to the results presented here.

I think it is extremely speculative to assume that changes in a modeled LAI estimate are proportional to % mortality, or total necromass. Virtually all allometric equations for biomass are nonlinear. A lot of hard work has been done in this area. See Marra et al. 2016 Biogeosciences. A number of papers have shown that non-linear size responses occur with common drivers of tree mortality. Droughts are thought to disproportionately kill large trees (Nepstad et al 2007 Ecology). The same goes for wind mortality (Rifai et al 2016 Ecological Applications), fire & wind (Silvério et al 2018 Journal of

Ecology). But otherwise there are so many other drivers of mortality that cannot simply be linearly scaled by height and LAI.

> Reply 18: Thank you, we agree. Our study confirms your statement, as we also see no linear trend between tree mortality and biomass mortality. This is true for many forest properties (LAI, forest height, biomass, GPP, NPP), as shown in Fig. 3 and Fig. 5.
> Also, we agree that changes in LAI are not enough for estimating mortality. Considering forest succession, we show that the proportion of biomass loss due to tree mortality can be estimated using a multiple linear regression model with LAI and forest height as proxy variables. We will revise the text to point this out.
> For our replies to the comments on the quality of our forest model, please see our reply 4, for that on the acknowledgment of previous research, please see our reply 7.
> We would like to note that in our study we use the term "biomass mortality" and by this we mean the proportion of the annually dying aboveground biomass (wood + leaves). This does not include deadwood pools and litter, as is sometimes the case in studies on "necromass" of tropical forests (Palace et al., 2012).

19) Barlow et al 2003 Ecology Letters; Fauset et al., 2019 Frontiers in Earth Science; Fisher et al., 2009 Ecology Letters; Chambers et al., 2009 Ecology Letters; Chambers et al 2013 PNAS; Marra et al 2014 PlosOne; Marra et al., 2018 Global Change Biology; McDowell et al 2018 New Phytologist; Negrón-Juarez et al., 2018; Negrón-Juarez et al., 2010 Geophysical Research Letters; Rifai et al., 2016 Ecological Applications; Sivério et al 2019 Journal of Ecology; and many many more.

> Reply 19: Thank you for the literature, we will include it in the text. For the full reply to this comment, please see our reply 7.

20) The x & y axes on figure 6 should be flipped in my opinion. There are some countering opinions on this, but typically observations are on the y-axis. However, there appears to be some non-linear influence between the quasi-observations and simulations of biomass mortality (necromass) that is not (or perhaps cannot be) captured with the linear regression.

> Reply 20: Thank you, we will flip the x-axis and y-axis of Fig. 6.
> We agree with the reviewer that not all forest stands can be described by the statistical model. There is a small number of stands with observed low mortality but predicted high mortality. These are forests of the simulated age 0 – 10 years. We will look at the properties of these forest stands to learn from them for the statistical model (Fig. 6 and Fig. S5).
> We think, linear regression is a suitable method for the data because the residuals are normally distributed around the expectation value (see Fig. S2b) and the remaining trend in the residuals is almost 0.

21) Scaling from one forest plot to a large region: 1) Again, it should be made absolutely clear, repeatedly and throughout the abstract that these results are based on model simulations. This includes the estimates of LAI, which are indeed modeled and not directly observed. 2) The arguments of this manuscript seem to be heavily dependent upon the veracity of the MCD15 product - however any optical remote sensing product has saturation effects when forest canopies are dense with an LAI > 4. 3) The assumption of the Paracou forest plot being representative of the entirety of French Guiana is exceptionally misplaced. The climate of Paracou is influenced by its proximity to the Atlantic. The supplemental figure S3, for example shows two aspects of how this site cannot represent the entirety of French Guiana. No one site can really be claimed to be representative of such a large area. 4) The simulations do not appear to be very realistic. Paracou exists upon relatively infertile soil with extremely limited Phosphorus. The simulation of approaching 500 Mg Biomass/Ha in less than 50

years is inconceivable with field measurements of NPP. These numbers should be compared with field observations in around the tropical forests of the Guiana shield.

Reply 21: Thank you for pointing this out.
@ 1. We will present more clearly the results, which are based on a fusion of simulation results and remote sensing measurements.
@ 2. Thank you for pointing out the limitations of LAI estimation with MODIS (saturation effect). We will clearly mention this in the revision. To be able to quantify the consequences of this limitation on our results, we will perform a LAI sensitivity analysis (see also our reply 16).
@ 3. Our model simulations represent the dynamics of Terra Firme forests, which cover 95 % of the forested area in French Guiana (Hammond, 2005; Stach et al., 2009). Our simulations follow the tradition of global vegetation models, which use similar assumptions. However, we include considerably more PFTs and structural variability. We will add this important point in the manuscript.
@ 4. We will test our model results with available data sets for FG (see reply 4).

22) Data availability: The value of this model focused manuscript is markedly reduced if the data and code are not openly available. I think the unavailability of the data and code is also contrary to the journal guidelines (https://www.biogeosciences.net/about/data_policy.html). If the data is available, then make it available - otherwise a detailed statement is required as to why it is not available. The need to contact the authors is especially burdensome upon the reader and is unlikely to be robust against the effects of time. Can the authors really guarantee they will always be around to provide the data and code when requested? Finally, even if FORMIND is available through other means, the results of this manuscript are unreproducible if it is not specific to the exact variant of FORMIND used in this manuscript.

Reply 22: Thank you. We agree with the reviewer and very much support free access to our data and model. For the full reply to this comment, please see our reply 6.

Line Comments

Figure 1 caption: What is meant by rejuvenation?

We mean the establishment of young trees with DBH ≥ 0.1 m. We would change the term.

Figure 3 caption: Put parentheses around acronyms that are being defined.

Thank you, will be done.

Figure 7: Is the histogram meant to serve as a color bar for the left panel? This is not very clear if so. It would be better to add a color bar indicating what the color gradient signifies.

Thank you, will be done.

Table 2: Report the intercept value of the linear regression.

Thank you, will be done.

20: Multivariate regression is when there are multiple response variables in the same regression model. Perhaps 'multiple linear regression' is meant?

Thank you. We will change this throughout the text.

25: I cannot tell if this is in reference to a model simulation or field observations?

The sentence refers to our simulation results. We will reformulate the text.

38: The Pan 2011 estimate of 2.8 is on the higher end and was assembled, more or less, haphazardly from the available forest census data and country level reports. More recent estimates are available.

Thank you, we will find a more recent reference.

41: You might see Korner 2003 Science, Chambers et al 2013 PNAS, and Fisher et al 2008 Ecology Letters.

Thank you. We will consider these references.

67-68: I don't see how this statement can be justified.

Thank you. FORMIND describes the vertical and horizontal forest structure and forest dynamics in 20 m x 20 m patches (Fischer et al., 2016). The simulation area consists of such interacting patches and it can vary from one hectare to multiple km$^2$ (here: 16 ha). The patches interact by tree falling. Such mosaic of patches leads to heterogeneity in space. We will revise the text to clearly point this out.

81-83: I find this hard to justify. See comparisons on Paracou and Nouragues.

Thank you. We will revise this sentence.

86-87: Competition for water is a major axis not mentioned. The Guiana shield has been struck numerous times by severe drought effects.

Thank you.
We will reformulate the sentence to be more precise.

95: About the regression model, what is the response and what is the covariate?

We propose to label the axes in the diagram on biomass mortality with "observed" and "estimated" (see point 3 in Fig. 6).

105-110: The paper using FORMIND v3.2 appears to be focused upon estimation of biomass with respect to changes in forest management. The parameterized version of the model does not appear to be available from that publication either.

Thank you. For the reply to this comment, please see our reply 6.

119-120: I find the model simulating mortality with arbitrarily distributed spatial patterns to be extremely implausible. Wind, fire, floods _ these all have a distinct spatial component. This spatial component has implications for who dies, and the post-disturbance light environment. Moreover, disturbance in reality is a punctuated event. If I read the section 2.2.2 correctly - the imposed disturbance intensity is actually just a multiplier on the baseline mortality rate. I don't think this is really anywhere near representative of disturbance in tropical forests.

Thank you, we agree with the reviewer. Our study does not focus on short-term disturbances (e.g. fire, wind-throw, drought), but rather on the effects of long-term changes in the intensity of tree mortality. We will present this more clearly in the manuscript.

160: This does not make sense to me. Allometric equations for biomass are typically nonlinear. See the widely used models including height in Chave et al., 2014 Global Change Biology.

Thank you, we agree. In our forest model, we use a non-linear relationship between aboveground biomass ($B$) of a tree t and its stem diameter ($D$):
$$B_t = \frac{\pi}{4} \cdot D_t^2 \cdot H_t \cdot F_t \cdot \frac{\rho_t}{\sigma_t},$$
where $H$ is the tree height, $F$ is a form factor, $\rho$ is wood density and $\sigma$ is the fraction of aboveground biomass attributed to the stem (Fischer et al., 2016). Then, tree biomass is summed up to forest biomass. We will add this equation to the manuscript.

175-180: I don't understand what exactly was done here. Tab S1 in the supplement is actually a paragraph. Eq 5 and 6 appear to be the same equation. Was mortality derived from the simulation model? If this was the case, I don't think there is anything that can really justify this. The manuscript appears to be about upscaling mortality with remote sensing data - but the core critical part, the mortality - is derived from a simulation model. This is making a very large number of assumptions, which I find implausible.

Thank you. We will change the designation 'Tab' and delete equation 6.
For the reply to the comment about upscaling biomass mortality, please see our reply 4.

259: Why 2km2 when the coarsest RS data was 1km2?

Thank you. A coarser scale results in fewer uncertainties being reflected in the map by the regression model. One could scale down the resolution. We suggest reformulating the paragraph in the text. For the full reply to this comment, please see our reply 17.

340-345: I think comparison with field-based estimates of tau is important. I suggest reading more into the actual forest census-based literature to come up with more comparisons. The Erb 2016 & Carvalhais 2014 papers are focused upon simulation results, and I don't agree that 20-30 years is similar to 40 years.

Thank you. We well implement numbers based on literature from empirical studies. Also, we will reformulate the text regarding the numbers published by Erb et al. (2016) and Carvalhais et al. (2014).

390: Terra Firme (not Terra Firma) is more commonly used to refer to this type of tropical forest

Thank you, we will change the term throughout the text.

390: 'successional' -> 'succession'

Thank you, we will change the term.

**Literature**

Brienen, R. J. W., Phillips, O. L., Feldpausch, T. R., Gloor, E., Baker, T. R., Lloyd, J., Lopez-Gonzalez, G., Monteagudo-Mendoza, A., Malhi, Y., Lewis, S. L., Vásquez Martinez, R., Alexiades, M., Álvarez Dávila, E., Alvarez-Loayza, P., Andrade, A., Aragão, L. E. O. C., Araujo-Murakami, A., Arets, E. J. M. M., Arroyo, L., … Zagt, R. J. (2015). Long-term decline of the Amazon carbon sink. *Nature*, *519*(7543), 344–348. https://doi.org/10.1038/nature14283

Carvalhais, N., Forkel, M., Khomik, M., Bellarby, J., Jung, M., Migliavacca, M., Mu, M., Saatchi, S., Santoro, M., Thurner, M., Weber, U., Ahrens, B., Beer, C., Cescatti, A., Randerson, J. T., & Reichstein, M. (2014). Global covariation of carbon turnover times with climate in terrestrial ecosystems. *Nature*, *514*(7521), 213–217. https://doi.org/10.1038/nature13731

Erb, K.-H., Fetzel, T., Plutzar, C., Kastner, T., Lauk, C., Mayer, A., Niedertscheider, M., Körner, C., & Haberl, H. (2016). Biomass turnover time in terrestrial ecosystems halved by land use. *Nature Geoscience*, *9*(9), 674–678. https://doi.org/10.1038/ngeo2782

Fischer, R., Bohn, F., Dantas de Paula, M., Dislich, C., Groeneveld, J., Gutiérrez, A. G., Kazmierczak, M., Knapp, N., Lehmann, S., Paulick, S., Pütz, S., Rödig, E., Taubert, F., Köhler, P., & Huth, A. (2016). Lessons learned from applying a forest gap model to understand ecosystem and carbon dynamics of complex tropical forests. *Ecological Modelling*, *326*, 124–133. https://doi.org/10.1016/j.ecolmodel.2015.11.018

Hammond, D. S. (2005). Tropical forests of the Guiana Shield: Ancient forests of the modern world. In *Tropical Forests of the Guiana Shield: Ancient Forests in a Modern World*. CABI Publishing. https://doi.org/10.1079/9780851995366.0000

Hiltner, U., Bräuning, A., Huth, A., Fischer, R., & Hérault, B. (2018). Simulation of succession in a neotropical forest: High selective logging intensities prolong the recovery times of ecosystem functions. In *Forest Ecology and Management*. https://www.sciencedirect.com/science/article/pii/S0378112718311964

Myneni, R., Knyazikhin, Y., & Park, T. (2015). *MODIS/Terra+Aqua Leaf Area Index/FPAR 8-day L4 Global 500m SIN Grid V006*. NASA EOSDIS Land Processes DAAC. https://doi.org/doi.org/10.5067/MODIS/MCD15A2H.006

Palace, M., Keller, M., Hurtt, G., & Frolking, S. (2012). A Review of Above Ground Necromass in Tropical Forests. In *Tropical Forests*. InTech. https://doi.org/10.5772/33085

Simard, M., Pinto, N., Fisher, J. B., & Baccini, A. (2011). Mapping forest canopy height globally with spaceborne lidar. *Journal of Geophysical Research*, *116*(G4), G04021. https://doi.org/10.1029/2011JG001708

Soong, J. L., Janssens, I. A., Grau, O., Margalef, O., Stahl, C., Van Langenhove, L., Urbina, I., Chave, J., Dourdain, A., Ferry, B., Freycon, V., Herault, B., Sardans, J., Peñuelas, J., & Verbruggen, E. (2020). Soil properties explain tree growth and mortality, but not biomass, across phosphorus-depleted tropical forests. *Scientific Reports*, *10*(1), 1–13. https://doi.org/10.1038/s41598-020-58913-8

Stach, N., Salvado, A., Petit, M., Faure, J. F., Durieux, L., Corbane, C., Joubert, P., Lasselin, D., &

Deshayes, M. (2009). Land use monitoring by remote sensing in tropical forest areas in support of the Kyoto Protocol: the case of French Guiana. *International Journal of Remote Sensing*, *30*(19), 5133–5149. https://doi.org/10.1080/01431160903022969

---

## Author Comment (AC2) · 15 Dec 2020

**Reply on the revision 1 of the Manuscript BG-2020-264**

**Reviewer 2**

**Overview**

This interesting study uses a forest model to explore how biomass mortality rate varies as a function of changes in stem mortality rate for a tropical forest location in French Guiana. It then uses these simulations to create a simple emulator linking indicators of successional stage and the resulting rate of biomass mortality. The emulator is then used to estimate biomass mortality rates across the whole of the country. The model has been previously been evaluated at the reference site used in this region. Overall, I find this a novel approach to investigate spatial variations in biomass mortality rates as a result of differences in forest age. Such efforts are important to provide baseline levels of mortality against which future changes can be compared, as well as to provide insights into the mechanisms driving mortality rates and any associated trends. I would like to see the manuscript published, but prior to that there are several aspects that I think should be clarified or expanded upon, including some additional analysis to identify the robustness of the results.

**Main comments**

> Thank you very much for your helpful constructive comments. Below you will find our replies to your comments (highlighted in blue).

1. A stand-level relationship between height, LAI and biomass mortality rate, is being used to scale up across a broad geographical region. Forest height here is almost purely an indicator of age of the largest trees, since there is relatively little difference between the disturbance scenarios at equilibrium. LAI appears to be both an indicator of age and composition. Between them they appear to characterise well how biomass mortality changes over the successional sequence. But when using this relationship to scale up, what happens if resource availability is not constant over the region being scaled over? Different levels of resource availability may also influence height and LAI - I should imagine particularly in the equilibrium stage for height and LAI from ca. 80 years onwards. Does the derived relationship in Fig. 6 hold across a productivity gradient? This is touched on in the limitations discussed in section 4.5, but I think it really needs to be tested (and presumably would be relatively straightforward to do). Even if the productivity gradient across French Guiana is small (as effectively assumed on L103), I think it is important for readers to know how robust the relationship and method are for application to more diverse regions.

> Reply 1: Thank you for this comment. In our study, productivity varies across different successional states and across scenarios of different disturbance intensity (see Fig. S1). We believe there is a productivity gradient already accounted for.
> However, we will conduct additional simulations in which we investigate the influence of a broader productivity gradient (by varying tree photosynthesis rates) on forest biomass mortality. The results will be reported in the paper.
> We will be happy to include this interesting aspect in the discussion.

2. In a similar vein, 1) is it appropriate to liken the increase in biomass mortality rate with forest height in this study (driven by a uniform mortality rate change) to the increase in stem mortality rate with individual height in Stovall et al. (2019) (L306)? I think the mechanisms are quite different. Biomass mortality rate would be expected to increase as forests approach equilibrium biomass, as the size of biomass losses must start to approach that of biomass growth. But this does not have to imply that stem mortality rates increase with tree size - it could simply be that the trees that are

dying are typically larger. This is distinct from a mechanism in which individual tree mortality rates scale with individual height (e.g. Holzwarth et al., 2013; Rowland et al., 2015; Stovall et al., 2019). I suggest to add a bit more nuance in the discussion of this point. 2) As an alternative comparison, in section 4.2, can the regression slopes instead be linked to the biogeographical patterns for the wider region from Johnson et al. (2016)? These patterns have been linked to a gradient in disturbance intensity and whilst Johnson et al. present biomass and stem mortality, rather than height and biomass mortality, FORMIND is simulating all components, facilitating a comparison.

> Reply 2: Thank you for this comment.
> @ 1: We agree with you and we will add your helpful thoughts to the discussion section.
> @ 2: Thank you for the literature, which we will take up in the discussion.
> Johnson et al. (2016) investigate forests of the entire Amazon region. Without conducting further research, the regression model should not be applied to other forest types (other than Terra Firme forests). We will provide additional extrapolations using our regression model on remote sensing products from LAI and forest height of the Guiana Shield (outside French Guiana). Then, we will compare our results of biomass mortality with values from the study of Johnson et al. (2016).

3. It would also be good to see some independent evaluation of the extrapolation. Whilst observations for biomass mortality in the region are likely rather hard to come by, how similar is FORMIND simulated height and LAI to the Simard et al. and MODIS data used for the extrapolation? Are they very close to each other, or is a correction factor needed to account for biases in one or the other? I wonder if you could also compare biomass mortality rate with that from other plots in the Guiana Shield provided in Brienen et al. (2015)?

> Reply 3: Thank you very much. No correction factors were required for the extrapolation. A comparison shows the density distributions of both data sets of LAI and forest height with the ranges of FORMIND's simulation results (see figure 1). We will include such information in the results part and discuss their implications.

[Figure]

Figure 1: Density distributions of remotely sensed forest height (left panel) and LAI (right panel) of French Guiana. FORMIND simulations are distributed in similar ranges (green area).In order to show the quality of the country-wide biomass mortality estimations, we will compare the map with empirical data from different empirical sources (e.g., Brienen et al., 2015; Soong et al., 2020).

4. In section 4.4 it is stated that the new framework allows to assess residence time as a functional of successional stage, but I think this is a bit misleading. The term residence time comes loaded with implications about how long carbon stays in the system. But in a transient system (as opposed to an equilibrium one), this does not hold for the kind of calculation used here, and during succession the

deviation from equilibrium is quite marked. The mean time a molecule of C entering a 50-year-old forest remains in that forest will likely be very different to the reciprocal of the biomass mortality rate at 50 years, because that molecule is more likely than its predecessors to be incorporated into a longer-lived later-successional PFT. In comparison, the biomass mortality rate is equivalently normalised to biomass but does not come with the same implications – it is unambiguously the rate at which carbon is currently leaving the system at that moment. I suggest only to use the concept of residence time here when averaging over the whole region (and then to term it turnover time, following Sierra et al., 2017).

Reply 4: Thank you for pointing this out. We agree with the reviewer's suggestion and will change the terminology of the term to "turnover time".

5. The result that net primary production remained stable is very interesting and neat. But can you add a bit more discussion about what this result is ultimately based on? To what extent is it an emergent outcome of the model, versus an assumption that went into the PFT parameterisation?

Reply 5: Thank you. We will take up the reviewer's suggestion of broadening the discussion on stable NPP. Although NPP in the old-growth phase is comparable across all scenarios, the functional composition of PFTs differs. We will study the proportion of pioneers and the mean longevity of trees more intensively as this seems to be the explanation behind the phenomenon.

6. Equation 3 implies that GPP = AGB_dead, which cannot be the case, as autotrophic respiration, allocation to soft tissue and allocation belowground need to be subtracted from GPP in order to get to woody NPP (i.e. woody biomass increment), which would be considered equivalent to AGB_dead at equilibrium (assuming that AGB is only counting the woody component of the total biomass). So, the tau obtained from AGB_total/GPP would be much smaller than that from AGB_total/AGB_dead. As FORMIND simulates GPP, a turnover time metric for comparison with Carvalhais et al. (2014) could be calculated, but it should be defined separately to the biomass turnover time with respect to mortality.

Reply 6: Thank you for pointing this out. We will correct the derivations of the equations concerning turnover time.

7. The LAI and height products used for extrapolation have errors associated with them. To what extent do these errors propagate through to uncertainty in the biomass mortality rates? I think Fig. 7 should be associated with an error field at least based on the input uncertainty, if not also the uncertainty in the regression fit.

Reply 7: Thank you. We will perform an uncertainty analysis to evaluate (1) the impact of the errors associated with the remote sensing products used and (2) the errors related to the linear regression model. The comparison of the map with empirical data (see our reply 3) will also contribute to the error assessment.

8. Why only 1 month of LAI data (L170)? Doesn't this expose your results to potential seasonal LAI variations? Wouldn't an average over several years provide a more like for- like comparison with the model output?

Thank you for the hint that LAI is very volatile. We will incorporate the mean of the MODIS LAI data by using a longer data series (from the same month as the ICESat data used for Simard's forest height map). We will also investigate the sensitivity of our workflow concerning variations in LAI (see reply 1).

Minor comments

line 33. Imprecise statement. 471 Pg C is much less than half of the terrestrial carbon stock (assuming vegetation + soils) given most estimates. Given the reference to Pan et al. (2011), I think you mean "forest carbon stocks"?

> Thank you. We will correct it in the text.

l47. What is an, "increase in associated physiological mechanisms"?

> We mean an expected higher frequency and intensity of mortality drivers may result in an increase in tree mortality due to related physiological mechanisms and interactions that may underlie increasing tree mortality in tropical forests (McDowell et al., 2018). We will revise the sentence.

l51. I always find disturbance a slippery term which can be used to refer to a very wide range of things. In this paragraph you give a general list of things that can be referred to as disturbances. I think you are defining disturbance as everything which is not related to competition, which is fine of course, but could you give an explicit definition of what is considered as disturbance for the purposes of this study?

> Yes, we would introduce a definition of the term "disturbance" as follows: long-term increase in tree mortality which is not stand-replacing.

L191. GPP is not the right indicator for a statement about "fixing five times more carbon in biomass" (see above). Can you show woody NPP? Or simply say, "fixing five times more carbon"?

> Thank you. Yes, we will add woody NPP.

Fig. 4. ODM is not included in the figure, just the caption.

> ODM appears in the y-axis labeling as part of the mass unit of GPP and NPP.

L301. The statement needs refining. Tree height is the strongest predictor of tree mortality out of which basket of indicators? At the individual tree level other predictors can be very important (see e.g. Esquivel-Muelbert et al., 2020), so it needs to be clear what is being compared to what.

> The statement refers to the two proxy variables LAI and forest height, which are included in the regression model (Eq. 5).
> Besides LAI and forest height we pre-tested GPP, NPP, and biomass as possible proxy variables in different combinations (with and without interaction terms, with and without interceptions). We will add this information in the text.

L366. I would suggest that biomass mortality rates depended on functional composition and level of divergence of C input and output fluxes from equilibrium, with LAI and forest height being indicators of these, not the drivers themselves.

> Thank you. We agree and would paraphrase the text.

L396. "more biomass is dying", or, "biomass is dying at a faster rate"? (because the map shows rates, rather than fluxes).

Thank you for the hint. We would reformulate the text.

References suggested by Reviewer 2

Brienen, R. J. W., Phillips, O. L., Feldpausch, T. R., Gloor, E., Baker, T. R., Lloyd, J., Lopez-Gonzalez, G., Monteagudo Mendoza, A., Malhi, Y., Lewis, S. L., Vásquez Martinez, R., Alexiades, M., Álvarez Dávila, E. A., Alvarez-Loayza, P., Andrade, A., Aragão, L. E. O. C., Araujo Murakami, A., Arets, E. J. M. M., Arroyo, L., Aymard C., G. A., Bánki, O. S., Baraloto, C., Barroso, J. G., Bonal, D., Boot, R., Camargo, J. L., Castilho, C., Chama, V., Chao, K.-J., Chave, J., Comiskey, J. A., Cornejo, F., Da Costa, L., De Oliveira, E. A., Di Fiore, A., Erwin, T. L., Fauset, S., Forsthofer, M., Grahame, S. E., Groot, N. E., Herault, B., Higuchi, N., Honorio C., E., Keeling, H., Killeen, T., Laurance, W., Laurance, S., Licona, J.-C., Magnussen, W. E., Marimon, B. S., Marimon-Junior, B. H., Mendoza, C., Neill, D., Nogueira, E. M., Nunez, P., Pallqui Camacho, N. C., Parada, A., Pardo, G., Peacock, J., Pena-Claros, M., Pickavance, G. C., Pitman, N. C. A., Poorter, L., Prieto, A., Quesada, C. A., Ramírez, F., Ramírez- Angulo, H., Restrepo, Z., Roopsind, A., Rudas, A., Salomão, R. P., Schwarz, M., Silva, N., Silva-Espejo, J. E., Silveira, M., Stropp, J., Talbot, J., Ter Steege, H., Teran-Aguilar, J., Terborgh, J., Thomas-Caesar, R., Toledo, M., Torello-Raventos, M., Umetsu, R. K., Van Der Heijden, G. M. F., Van Der Hout, P., Guimaraes Vieira, I., Vieira, S. A., Vilanova, E., Vos, V. and Zagt, R. J.: Long-term decline of the Amazon carbon sink, Nature, 519(7543), 344–348, doi:10.1038/nature14283, 2015.

Esquivel-Muelbert, A., Phillips, O. L., Brienen, R. J. W., Fauset, S., Sullivan, M. J. P., Baker, T. R., Chao, K. J., Feldpausch, T. R., Gloor, E., Higuchi, N., Houwing-Duistermaat, J., Lloyd, J., Liu, H., Malhi, Y., Marimon, B., Marimon Junior, B. H., Monteagudo-Mendoza, A., Poorter, L., Silveira, M., Torre, E. V., Dávila, E. A., del Aguila Pasquel, J., Almeida, E., Loayza, P. A., Andrade, A., Aragão, L. E. O. C., Araujo-Murakami, A., Arets, E., Arroyo, L., Aymard C, G. A., Baisie, M., Baraloto, C., Camargo, P. B., Barroso, J., Blanc, L., Bonal, D., Bongers, F., Boot, R., Brown, F., Burban, B., Camargo, J. L., Castro, W., Moscoso, V. C., Chave, J., Comiskey, J., Valverde, F. C., da Costa, A. L., Cardozo, N. D., Di Fiore, A., Dourdain, A., Erwin, T., Llampazo, G. F., Vieira, I. C. G., Herrera, R., Honorio Coronado, E., Huamantupa-Chuquimaco, I., Jimenez-Rojas, E., Killeen, T., Laurance, S., Laurance, W., Levesley, A., Lewis, S. L., Ladvocat, K. L. L. M., Lopez-Gonzalez, G., Lovejoy, T., Meir, P., Mendoza, C., Morandi, P., Neill, D., Nogueira Lima, A. J., Vargas, P. N., de Oliveira, E. A., Camacho, N. P., Pardo, G., Peacock, J., Peña-Claros, M., Peñuela-Mora, M. C., Pickavance, G., Pipoly, J., Pitman, N., Prieto, A., Pugh, T. A. M., Quesada, C., Ramirez-Angulo, H., de Almeida Reis, S. M., Rejou-Machain, M., Correa, Z. R., Bayona, L. R., Rudas, A., Salomão, R., Serrano, J., Espejo, J. S., Silva, N., Singh, J., Stahl, C., Stropp, J., Swamy, V., Talbot, J., ter Steege, H., et al.: Tree mode of death and mortality risk factors across Amazon forests, Nat. Commun., 11(1), doi:10.1038/s41467-020-18996-3, 2020.

Holzwarth, F., Kahl, A., Bauhus, J. and Wirth, C.: Many ways to die – partitioning tree mortality dynamics in a near-natural mixed deciduous forest, J. Ecol., 101, 220–230, doi:10.1111/1365-2745.12015, 2013.

Johnson, M. O., Galbraith, D., Gloor, E., H, D. D., Guimberteau, M., Rammig, A., Thonicke, K., Verbeeck, H., Monteagudo, A., Phillips, O. L., Brienen, R. J.W., Feldpausch, T. R., G, L. G., Fauset, S., Quesada, C. A., Christoffersen, B., Ciais, P., Gilvan, S., Kruijt, B., Meir, P., Moorcroft, P., Zhang, K., Alvarez, E. A., Amaral, I., Andrade, A., Aragao, L. E. O. C., Arets, E. J. M. M., Arroyo, L., Aymard, G. A., Baraloto, C., Barroso, J., Bonal, D., Boot, R., Camargo, J., Chave, J., F, C. V., Ferreira, L., Higuchi, N. and Honorio, E.: Variation in stem mortality rates determines patterns of aboveground biomass in Amazonian forests: implications for dynamic global vegetation models, Glob. Chang. Biol., 22(12), 3996–4013, doi:10.1111/gcb.13315, 2016.

Pan, Y., Birdsey, R. a, Fang, J., Houghton, R., Kauppi, P. E., Kurz, W. a, Phillips, O. L., Shvidenko, A., Lewis, S. L., Canadell, J. G., Ciais, P., Jackson, R. B., Pacala, S. W., McGuire, a D., Piao, S., Rautiainen, A., Sitch, S. and Hayes, D.: A large and persistent carbon sink in the world's forests., Science (80-. )., 333(6045), 988–93, doi:10.1126/science.1201609, 2011.

Rowland, L., da Costa, A. C. L., Galbraith, D. R., Oliveira, R. S., Binks, O. J., Oliveira, A. A. R., Pullen, A. M., Doughty, C. E., Metcalfe, D. B., Vasconcelos, S. S., Ferreira, L. V, Malhi, Y., Grace, J., Mencuccini, M. and Meir, P.: Death from drought in tropical forests is triggered by hydraulics not carbon starvation., Nature, 528(7580), 119–122, doi:10.1038/nature15539, 2015.

Sierra, C. A., Müller, M., Metzler, H., Manzoni, S. and Trumbore, S. E.: The muddle of ages, turnover, transit, and residence times in the carbon cycle, Glob. Chang. Biol., 23(5), 1763–1773, doi:10.1111/gcb.13556, 2017.

Stovall, A. E. L., Shugart, H. and Yang, X.: Tree height explains mortality risk during an intense drought, Nat. Commun., 10, 4385, doi:10.1038/s41467-019-12380-6, 2019.

**Literature**

Brienen, R. J. W., Phillips, O. L., Feldpausch, T. R., Gloor, E., Baker, T. R., Lloyd, J., Lopez-Gonzalez, G., Monteagudo-Mendoza, A., Malhi, Y., Lewis, S. L., Vásquez Martinez, R., Alexiades, M., Álvarez Dávila, E., Alvarez-Loayza, P., Andrade, A., Aragão, L. E. O. C., Araujo-Murakami, A., Arets, E. J. M. M., Arroyo, L., … Zagt, R. J. (2015). Long-term decline of the Amazon carbon sink. *Nature*, *519*(7543), 344–348. https://doi.org/10.1038/nature14283

Fischer, R., Bohn, F., Dantas de Paula, M., Dislich, C., Groeneveld, J., Gutiérrez, A. G., Kazmierczak, M., Knapp, N., Lehmann, S., Paulick, S., Pütz, S., Rödig, E., Taubert, F., Köhler, P., & Huth, A. (2016). Lessons learned from applying a forest gap model to understand ecosystem and carbon dynamics of complex tropical forests. *Ecological Modelling*, *326*, 124–133. https://doi.org/10.1016/j.ecolmodel.2015.11.018

Hammond, D. S. (2005). Tropical forests of the Guiana Shield: Ancient forests of the modern world. In *Tropical Forests of the Guiana Shield: Ancient Forests in a Modern World*. CABI Publishing. https://doi.org/10.1079/9780851995366.0000

Hiltner, U., Bräuning, A., Huth, A., Fischer, R., & Hérault, B. (2018). Simulation of succession in a neotropical forest: High selective logging intensities prolong the recovery times of ecosystem functions. In *Forest Ecology and Management*. https://www.sciencedirect.com/science/article/pii/S0378112718311964

Johnson, M. O., Galbraith, D., Gloor, M., De Deurwaerder, H., Guimberteau, M., Rammig, A., Thonicke, K., Verbeeck, H., von Randow, C., Monteagudo, A., Phillips, O. L., Brienen, R. J. W., Feldpausch, T. R., Lopez Gonzalez, G., Fauset, S., Quesada, C. A., Christoffersen, B., Ciais, P., Sampaio, G., … Baker, T. R. (2016). Variation in stem mortality rates determines patterns of above-ground biomass in Amazonian forests: implications for dynamic global vegetation models. *Global Change Biology*, *22*(12), 3996–4013. https://doi.org/10.1111/gcb.13315

McDowell, N., Allen, C. D., Anderson-Teixeira, K., Brando, P., Brienen, R., Chambers, J., Christoffersen, B., Davies, S., Doughty, C., Duque, A., Espirito-Santo, F., Fisher, R., Fontes, C. G., Galbraith, D., Goodsman, D., Grossiord, C., Hartmann, H., Holm, J., Johnson, D. J., … Xu, X. (2018). Drivers and mechanisms of tree mortality in moist tropical forests. In *New Phytologist* (Vol. 219, Issue 3, pp. 851–869). Blackwell Publishing Ltd. https://doi.org/10.1111/nph.15027

Myneni, R., Knyazikhin, Y., & Park, T. (2015). *MODIS/Terra+Aqua Leaf Area Index/FPAR 8-day L4 Global 500m SIN Grid V006*. NASA EOSDIS Land Processes DAAC. https://doi.org/doi.org/10.5067/MODIS/MCD15A2H.006

Palace, M., Keller, M., Hurtt, G., & Frolking, S. (2012). A Review of Above Ground Necromass in Tropical Forests. In *Tropical Forests*. InTech. https://doi.org/10.5772/33085

Simard, M., Pinto, N., Fisher, J. B., & Baccini, A. (2011). Mapping forest canopy height globally with spaceborne lidar. *Journal of Geophysical Research*, *116*(G4), G04021. https://doi.org/10.1029/2011JG001708

Soong, J. L., Janssens, I. A., Grau, O., Margalef, O., Stahl, C., Van Langenhove, L., Urbina, I., Chave, J., Dourdain, A., Ferry, B., Freycon, V., Herault, B., Sardans, J., Peñuelas, J., & Verbruggen, E. (2020).

Soil properties explain tree growth and mortality, but not biomass, across phosphorus-depleted tropical forests. *Scientific Reports*, *10*(1), 1–13. https://doi.org/10.1038/s41598-020-58913-8

Stach, N., Salvado, A., Petit, M., Faure, J. F., Durieux, L., Corbane, C., Joubert, P., Lasselin, D., & Deshayes, M. (2009). Land use monitoring by remote sensing in tropical forest areas in support of the Kyoto Protocol: the case of French Guiana. *International Journal of Remote Sensing*, *30*(19), 5133–5149. https://doi.org/10.1080/01431160903022969